# Understanding the Generalization Gap in Visual Reinforcement Learning

## Abstract

Deep Reinforcement Learning (RL) agents have achieved superhuman performance on several video game suites. However, unlike humans, the trained policies fail to transfer between related games or even between different levels of the same game. Recently several techniques have been proposed to close the generalization gap such as data augmentation, domain invariant feature learning, separation of actor and critic networks, etc. However, the transfer performance of RL agents still remains unsatisfactory. In this work, we conduct a large scale empirical study using procedurally generated video games to understand why the generalization gap still exists. We also show that simple auxiliary tasks can improve generalization of policies. Furthemore, contrary to the belief that adaptation to new levels requires finetuning all layers of the policy network, we find that features in the visual trunk can be kept fixed and only the parameters that use visual features to predict actions require finetuning. Finally, to inform fruitful avenues for future research, we construct simple oracle methods that close the generalization gap.

## 1 Introduction

Deep Reinforcement Learning (RL) has achieved tremendous success on several video game suites (Mnih et al., 2013; Vinyals et al., 2019; Berner et al., 2019). However, state-of-the-art (SOTA) RL agents are usually trained and tested on the same game. It turns out that the learned policy fails to transfer to different games or even unseen levels of the same game (Cobbe et al., 2018; 2019). In contrast, humans are remarkable at transferring to new tasks and environments. Behavior studies (Dubey et al., 2018) show that humans make extensive use of visual priors such as object permanence, sub-goals, and intuitive physics when learning new games. Without these priors, data efficiency of humans drops an order of magnitude (Dubey et al., 2018). (Sax et al., 2018) learns visual priors with supervised learning on datasets with manual annotations of several semantically meaningful features. In practice however, acquiring such datasets is challenging.

In the absence of labelled datasets and human supervision, past works have explored three main ways to incorporate visual priors to improve generalization. The first paradigm is *data augmentation* (Michael et al., 2020; Kostrikov et al., 2020) where invariance in the learned representations is induced by training the agent on a large set of task-irrelevant variations of the observed data. Since the distinction between what is relevant vs irrelevant is task-dependent, the optimal choice of data augmentation is also task dependent. To avoid manual selection of data augmentations, UCB-DrAC (Raileanu et al., 2020) used upper confidence bound (UCB) algorithm (Auer, 2002) for choosing optimal augmentations. However, it remains unclear whether choosing one out of many task-agnostic data augmentations, such as cropping, helps in closing the generalization gap. In this paper, we show that set of transformations used in state-of-the-art methods are *insufficient* at closing the generalization gap. On the other hand, we also show that task-informed data augmentations close the generalization gap, but require task variation information during training. How to construct task sensitive data augmentations remains an open question and an avenue of future research. We discuss this finding in Section 4.

The second paradigm for transfer learning is to leverage *domain confusion* (Tzeng et al., 2014; 2015; Hoffman et al., 2013), technique commonly used in computer vision to make the source data distribution indistinguishable from the target distribution. The main idea is to discourage learning of spurious domain/level-specific features by enforcing the constraint that it should not be possible

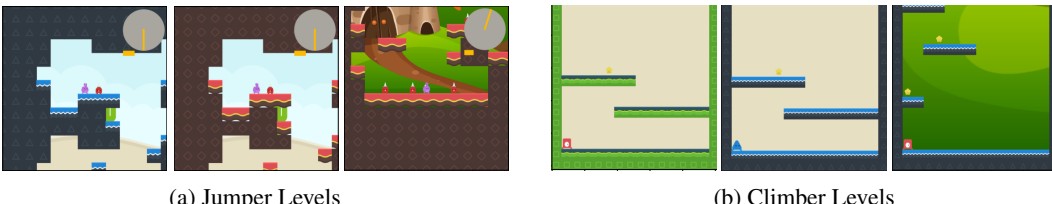

(a) Jumper Levels         (b) Climber Levels

Figure 1: Visualization of different levels of (a) Jumper and (b) Climber. In both the games, the first and second level differ in theme whereas the first and third level differ in both theme and layout.

to predict the identity of the domain/level from the policy features. iDAAC (Raileanu & Fergus, 2021) used this technique to improve policy generalization. However, our investigation reveals that even policies that generalize well contain domain/level specific information. In hindsight, this is not surprising. On image classification tasks, the identity of the object doesnot change based on factors such as background clutter, size, etc. However, small changes in the size of obstacles or platforms can drastically change the optimal policy. Therefore, the policy must encode information about level-specific game layout, making application of domain invariance to policy learning challenging. Details of this investigation are provided in Section 5.

The third paradigm of improving generalization involves using *auxiliary prediction tasks* (such as depth prediction, image reconstruction, contrastive learning) to regularize the policy features to prevent learning spurious features (Jaderberg et al., 2016; Srinivas et al., 2020). In this work, we choose the auxiliary task of inverse model prediction (Agrawal et al., 2016) due to its simplicity. This improves policy generalization. We detail this finding in Section 6.

It is well-known that choice of hyper parameters has a substantial impact on the performance of RL agents. When it comes to policy generalization, discussions around hyperparameter selection have been limited. We show that state-of-the-art (SOTA) methods that improve policy generalization require careful hyperparameter selection for each task. When these methods are constrained to choose one set of hyperparameters across tasks, they donot outperform the base PPO algorithm. We detail this finding in Section 7.

Finally, we probe if deep RL algorithms learn spurious features that deters performance of the same task in new environments. Our experiments reveal that visual features learned by training on a limited set of levels of one game donot require adaption to achieve good performance on new levels. Infact, only the layers of the policy network that convert visual features into actions require finetuning. Details of this investigation are provided in Section 8.

In summary, our experiments on Procgen (Cobbe et al., 2019), a suite of procedurally generated games reveal that (i) task-informed data augmentation closes the generalization gap; (ii) learning level-invariant (or domain-invariant) features is not necessary for good generalization; (iii) regularizing the policy features using simple auxiliary tasks can improve generalization; (iv) SOTA methods that improve policy generalization rely on careful hyperparameter selection; (v) training on a limited number of levels of the same game doesnot result in learning of level-specific spurious features. When operating on new levels, good performance can be achieved by only fine-tuning two layers that transform these features into actions.

## 2 RELATED WORK

Prior works (Cobbe et al., 2018; 2019; Justesen et al., 2018; Zhang et al., 2018a;b; Juliani et al., 2019; Rajeswaran et al., 2017; Raileanu & Rocktäschel, 2020; Grigsby & Qi, 2020; Kuttler et al., 2020; Farebrother et al., 2018) have established the problem of overfitting in reinforcement learning. In response, multiple methods have been proposed to mitigate overfitting. (Cobbe et al., 2018; 2019) showed that classical techniques such as dropout (Srivastava et al., 2014), L2 regularization, and batch normalization (Ioffe & Szegedy, 2015) – originally developed for supervised learning – reduce the generalization gap in RL. Other works (Michael et al., 2020; Cobbe et al., 2018; Ye et al., 2019; Raileanu et al., 2020; Wang et al., 2020) have made use of data augmentation to learn policies that generalize well. Another popular approach is to use representation learning techniques such as variational information bottleneck (Igl et al., 2019) and bisimulation metrics (Agarwal et al., 2021;

Zhang et al., 2020) to help with the problem of overfitting in RL. (Cobbe et al., 2020; Raileanu & Fergus, 2021) learned a separate network for policy and value function and showed that it helps reduce the generalization gap in RL. (Sax et al., 2018) showed that using mid-level visual features optimized for segmentation, depth, keypoints, surface normal prediction and others improve transfer. In our work, rather than proposing a novel approach to solve overfitting, we analyze the limitations of the current techniques and propose avanues for improvements.

## 3 PRELIMINARIES

We consider a distribution of Partially Observable Markov Decision Process (POMDPs) $p(\mathcal{M})$ such that $\mathcal{M}_i = (S_i, O_i, A_i, T_i, \Omega_i, R_i, \gamma) \sim p(\mathcal{M})$. We can think of different $\mathcal{M}_i$ as instances of same task in different environments. $S_i$ is the state space, $O_i$ is the observation space, $A_i$ is the action space, $T_i(s'|s, a)$ is the transition function, $\Omega_i(o'|s, a)$ is the observation function, $R_i(s, a)$ is the reward function and $\gamma$ is the discount factor. The goal is to find a policy $\pi_\theta$ that maximizes the expected sum of discounted rewards over the distribution of POMDPs, $J(\pi_\theta) = \mathbb{E}_{\pi, p(\mathcal{M})} \left[ \sum_{t=0}^{T-1} \gamma^t R_i(s_t, a_t) \right]$.

During training, we only have access to a limited number of POMDPs $\hat{\mathcal{M}} = \{\mathcal{M}_i\}_{i=1}^N$. Our goal is to find a policy $\pi_\theta$ that generalizes to new POMDPs sampled from $p(\mathcal{M})$.

### 3.1 ENVIRONMENT SETUP

We conduct experiments using the ProcGen suite (Cobbe et al., 2019), a collection of 16 procedurally generated video games. Each game contains multiple levels wherein the agent needs to perform the same task, but the environment varies due to changes in position of objects, their textures and the background texture. We refer to variation in levels resulting solely from changes in texture of object, agent and the background as *theme* variation. We collectively refer to variation in levels resulting from changes in positions of the objects as *layout* variation. Figure 1 illustrates theme and layout variation for two games, *Jumper* and *Climber*. For each game, the first and the second level differ in theme whereas the first and the third level differ in both the theme and the layout.

### 3.2 METHODS

We use standard neural network architecture to represent the policy, $a \sim \pi_\theta(a|s)$, consisting of: a visual encoder $z = \pi_{\theta_1}^{enc}(s)$ and a policy head $a \sim \pi_{\theta_2}^h(a|z)$ with parameters $\theta_1, \theta_2$ respectively. Collectively, they are referred as $\theta = [\theta_1, \theta_2]$. The architecture of the visual encoder is borrowed from IMPALA (Espeholt et al., 2018) and is a ResNet with 15 convolutional layers. It outputs a flattened feature map ($z$) that we refer to as the *visual feature*. These features are passed into a two 2-layer fully-connected neural networks with 256 hidden units for predicting the actions (i.e., the policy head) and the value function $\left( \text{i.e., the value head, } V_\psi^h(z) \right)$.

We analyze the generalization performance of PPO (Schulman et al., 2017) and three state-of-the-art (SOTA) methods:

- **UCB-DrAC (Raileanu et al., 2020)** uses Upper Confidence Bound (UCB) to automatically select game specific data augmentations out of eight possible ones: *crop*, *grayscale*, *cutout*, *cutout-color*, *flip*, *rotate*, *random convolution* and *color jitter*. For brevity, we sometimes refer to UCB-DrAC as simply DrAC.

- **DAAC (Raileanu & Fergus, 2021)** trains two separate networks for representing the policy and the value function. The intuition is that in POMDPs, the value network will overfit, but the policy can still generalize. The seperation of networks prevents interference in visual features learned for representing the value function and the policy. PPG (Cobbe et al., 2020) also trains separate networks for actor and critic but mainly focuses on improved training of policies and not their generalization performance.

- **iDAAC (Raileanu & Fergus, 2021)** adds an auxiliary loss to the training objective of DAAC to encourage domain invariance.

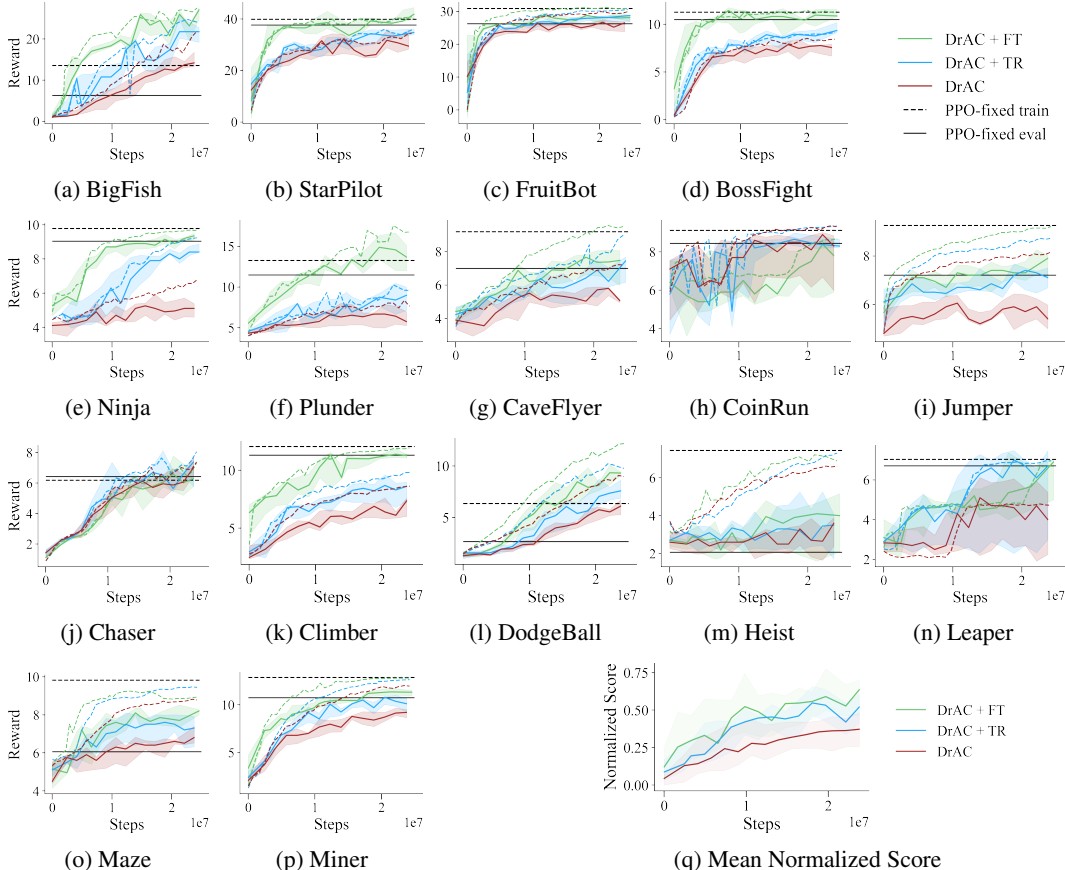

Figure 2: UCB-DrAC with theme randomization (DrAC-TR) gets close to the test performance of DrAC-FT on games with fixed themes indicating that it learned theme invariant features. Thus, it outperforms UCB-DrAC implying the need for better domain randomization and data augmentation. Black horizontal lines are the performance of a PPO baseline trained with fixed texture. Dashed line is training performance.

We use PPO's pytorch implementation (Kostrikov, 2018), the official implementation for UCB-DrAC (Raileanu et al., 2020), and the official implementation for DAAC/iDAAC (Raileanu & Fergus, 2021). See Appendix B for the hyperparameters used.

## 3.3 EVALUATION METRICS

Different procgen games have different score ranges which makes it difficult for us to average the scores across games. Recent works (Raileanu et al., 2020; Raileanu & Fergus, 2021) used percentage improvement over PPO for averaging across games. However, this metric doesn't properly take reward range of a game into account and is also biased by PPO's performance on that game, thereby being more susceptible to large variations. To remedy this, as suggested in (Cobbe et al., 2019), we use the *normalized score* $\frac{R - R_{min}}{R_{max} - R_{min}}$ to compute mean scores (MNS) across games. Here, $R_{max}$ and $R_{min}$ are game-dependent constants provided in (Cobbe et al., 2019). To calculate the final train and test performances of a policy, we evaluate the policy 100 times on train levels (1-200) and test levels for each game. We report the mean and standard deviation of final train and test performances computed across 5 different seeds. We train the policy on the first two hundred levels and evaluate it on a subset of remaining levels (i.e. approximately two million unique levels).

Table 1: Performance of level classifier trained on the encoder features of policies learned with PPO on 200 levels, with PPO on 100k levels, with DAAC on 200 levels, and with iDAAC on 200 levels, averaged across all procgen games.

|       | PPO (200)      | PPO (100k)     | DAAC           | iDAAC          |
|-------|----------------|----------------|----------------|----------------|
| Train | $0.95 \pm 0.1$ | $0.96 \pm 0.1$ | $0.95 \pm 0.1$ | $0.94 \pm 0.11$ |
| Test  | $0.91 \pm 0.16$ | $0.87 \pm 0.18$ | $0.89 \pm 0.12$ | $0.87 \pm 0.13$ |

## 4 INVESTIGATING POLICY GENERALIZATION WITH DATA AUGMENTATION

A policy can fail to generalize to a new level due to inability to deal with either: (i) *layout* variation or (ii) *theme* variation. To tackle the second issue, prior work (Raileanu et al., 2020; Michael et al., 2020) has employed data augmentation. If these methods are successful at achieving theme in-variance, then their performance on test-levels should be same as a policy that is trained and evaluated on ProcGen games with fixed themes. To test if this is true, we trained UCB-DrAC, a state-of-the-art data augmentation method on ProcGen games with fixed themes. We refer this to oracle method as DrAC+FT, where FT stands for *fixed theme*. Results in Figure 2 show a significant test performance difference between UCB-DrAC and DrAC+FT which suggests that UCB-DrAC does not achieve theme invariance. Moreover, as shown in Figure 3, UCB-DrAC does not outperform a well-tuned PPO baseline when averaged across all the games.

The above results raise a question: is it possible to close the generalization gap by adopting a better data augmentation scheme? To answer this, we constructed game levels that had a fixed layout, but varied only in theme. We refer to a PPO agent trained on this baseline as DrAC-TR for *theme randomization*. Our intent is to use this domain randomization scheme, that makes use of privileged access to the game engine, as a proxy of better data augmentation schemes. Figure 2 shows that DrAC+TR greatly improves over UCB-DrAC and gets closer to the test performance of DrAC+FT. Our result demonstrates the viability of pushing for stronger data augmentation schemes for producing theme invariance and thus policy generalization. We hope to encourage the development towards this venue of research with our reported findings.

## 5 INVESTIGATING POLICY GENERALIZATION WITH DOMAIN CONFUSION

Another way to learn generalizable policies is by using *domain confusion* (Tzeng et al., 2014; 2015; Hoffman et al., 2013) which discourages policies from predicting task irrelevant properties of the environment, thereby encouraging them to focus on task relevant properties. In addition to separating the policy and the value network, (Raileanu & Fergus, 2021) used this technique to prevent the policy from learning level-specific features and obtained SOTA results. Given these findings, we might think that we should make policies level invariant to obtain better generalization. However, a level (in a game) is not only defined by its theme but also by its layout which contains task relevant information. Therefore, learning level-invariant policies might remove layout features important for solving the task and thus hurt the performance of these policies. Hence, it's not clear if the policies, that generalize well, are independent of level-specific features.

To answer this, we first take the policies trained by PPO on 100k levels, by DAAC on 200 levels and by iDAAC on 200 levels on different games of procgen. All these policies have good generalization capabilities. We want to test if the visual features coming from these policies contain level-specific features. Therefore, we collect 1 trajectory with random exploration for each level from 1 to 200 (per game) and label each state in the trajectory with the corresponding level id. We then mix all the (state, level id) tuples in the collected trajectories and call the resulting dataset $\mathcal{D}$. The first $80\%$ of $\mathcal{D}$ is labelled as train set and the remaining as the test set. For each of the policies described above, we use the train set to learn a linear level classifier on top of their visual features and evaluate the performance of the learned classifier on the test set. Table 1 shows that each of the classifier learned obtained a good train and test performance averaged across all games. We further provide the performance of these classifiers on individual games in the Appendix. This implies that all the above mentioned policies, with good generalization capabilities, contain level-specific features. To further bolster our claim, we use the train set to learn a linear classifier on top of visual features of

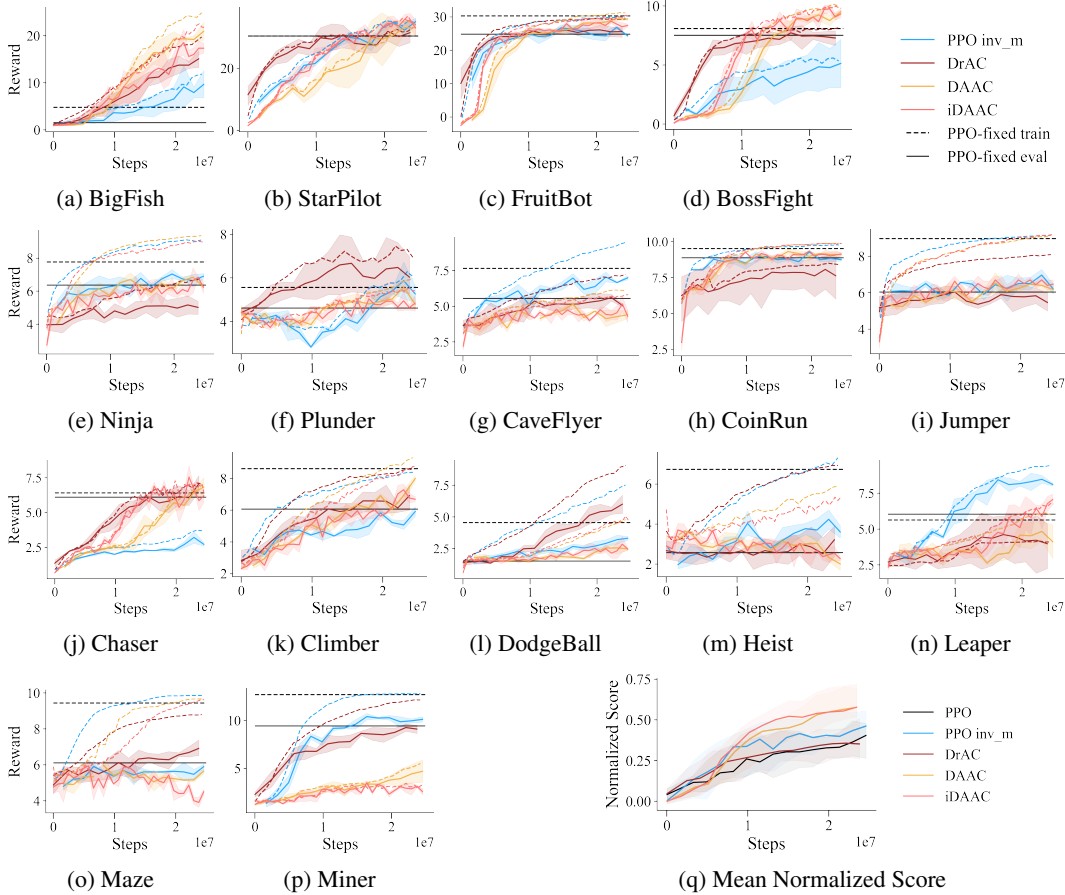

Figure 3: PPO with inverse model regularization (PPO inv m) is competitive (in terms of test performance) against UCB-DrAC, DAAC and iDAAC in the ProcGen domain despite being a much simpler method. Black horizontal lines are the test performance of a PPO baseline. Dashed line is training performance.

policies trained by PPO on 200 levels and evaluate its performance on the test set. We see that its performance is similar to that of the previous classifiers. This shows that policies have level specific features regardless of whether it generalizes or not.

DAAC/iDAAC (Raileanu & Fergus, 2021) performs well due to less overfitting as a result of separation of policy and value networks and the regularization of policy features from its auxiliary losses. However, the insight that these methods don't learn level-specific features and discouraging the policies to learn level-specific features leads to better generalization, isn't empirically grounded.

## 6 INVESTIGATING POLICY GENERALIZATION WITH AUXILIARY TASKS

Data augmentation and domain confusion can be thought of as two separate methods of constraining the learned visual features to potentially improve generalization. A third way to constrain features is to optimize them for *auxiliary* objectives in addition to the policy and value prediction losses. (Jaderberg et al., 2016) showed that depth prediction improves both data efficiency and the generalization of learned policies on first-person navigation tasks. However, depth is an additional sensory modality that is not always available.

Therefore, we propose a simpler auxiliary task of encouraging the visual features coming from two consecutive states to predict the corresponding action. This inverse model regularization has been shown to be an effective regularization technique in prior works (Agrawal et al., 2016; Pathak et al., 2017; 2018). Let $z_t = \pi_{\theta_1}^{enc}(s_t)$. If $f_\phi$ is the inverse model and $(s_t, a_t, s_{t+1})$ is the state action state

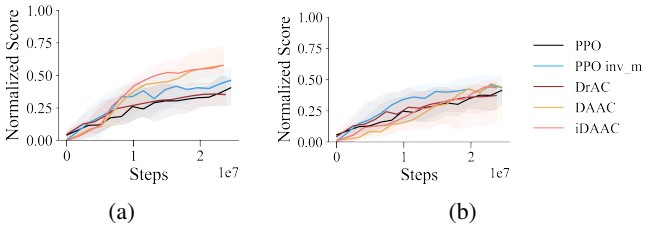

Figure 4: We compare PPO, PPO with inverse model regularization (PPO inv m), UCB-DrAC, DAAC, and iDAAC on test levels (a) when using different hyperparameters for each game and (b) when using a single set of hyperparameters across all games

tuple, then this regularization minimizes $\mathcal{CE}(a_t, f_\phi(z_t, z_{t+1}))$ where $\mathcal{CE}$ is the cross-entropy loss. Figure 3 shows that this regularization is competitive with UCB-DrAC in terms of test performance despite being simpler.

## 7 EFFECTS OF HYPERPARAMETER SELECTION ON POLICY GENERALIZATION

Until now we have investigated several methods for learning policies that generalize across levels. However, with exception of UCB-DrAC, we found that all the other methods (PPO inv m, DAAC, iDAAC) require careful hyperparameter selection to improve over the base PPO algorithm. To quantify the impact of careful hyperparameter selection, we chose a single set of hyperparameters for each method that maximizes the mean normalized score (MNS) evaluated on validation levels $201 - 400$. We then compare these policies on the test levels. Figure 4 shows that average test performance across games of all methods is similar to PPO. Additional details about hyperparameter selection are provided in Appendix B. These results suggest that improvements over PPO heavily depends on careful hyperparameter selection on each game. One could argue that different games require different hyperparameters due to their unique properties. However, careful hyperparameter tuning on each game is compute-intensive, and there's no guidance on how to set these hyperparameters other than performing a grid search. Hence, automated tuning of hyperparameter to save compute costs is an important avenue for future research.

## 8 INVESTIGATING POLICY ADAPTATION TO NEW LEVELS

Until now we have investigated zero-shot evaluation of policies on test levels. The other possibility is to finetune the policies on new levels.

First, we train a policy on a limited number of levels (i.e., levels $1 - 200$). However, rather than directly evaluating the trained policy, we finetune the trained policy on new levels (i.e., levels $201 - 400$) to achieve the train-time performance quickly. In this setting, it is natural to wonder if finetuning of the entire policy is required to achieve the train-time performance on new levels. To answer this question, we fix the visual encoder $\pi_{\theta_1}^{enc}$ of a policy trained using PPO on levels $1 - 200$ of a game and finetune only the policy head $\pi_{\theta_2}^h$ (as well as the value head $V_\psi$) on levels $201 - 400$ of the game. If finetuning the entire policy on levels $201 - 400$ is required to achieve the train-time performance, then finetuning only the policy head on those levels should lead to a sub-optimal performance.

Figure 5 shows that the average returns of the policy on levels $201 - 400$ become similar to the average returns of the policy on levels $1 - 200$ after finetuning the policy head. Furthermore, finetuning the entire policy on levels $201 - 400$ levels gives a similar average return. These observations imply that finetuning only the policy head (on levels $201 - 400$) is sufficient to recover the average return of levels $1 - 200$. Additionally, we see that end-to-end training of policy from scratch on levels $201 - 400$ is less sample-efficient than finetuning the policy head as well as finetuning the entire policy. Moreover, training of policy head with randomly initialized visual encoder as well as imagenet (Krizhevsky et al., 2012) trained visual encoder on levels $201 - 400$ leads to poor average returns. These observations show that the visual encoder features learned from levels $1 - 200$ are meaningful and help finetune both the policy head and the entire policy. Overall, this implies that the visual features learned on training levels ($1 - 200$) are meaningful and can be kept fixed when

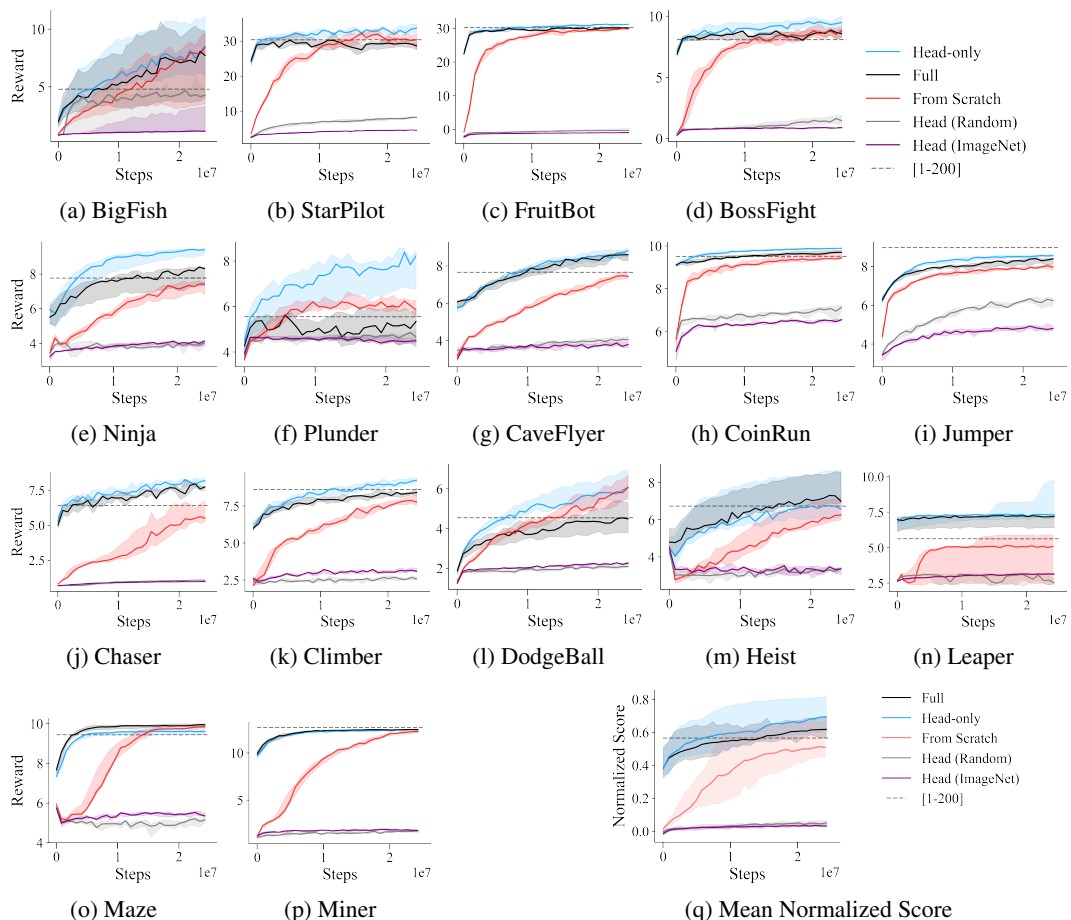

Figure 5: **Transfer**: We compare fine-tuning the policy head (Head-only) versus the entire policy network (Full) on new levels $(201 - 400)$. The policy is trained on 200 levels. Fine tuning just the policy head allows us to recover this performance on the training levels ([1 - 200]), showing that the visual features only require a small number of training levels to generalize. We additionally include the learning curve of a policy trained from scratch (From Scratch), a policy head trained on a randomly initialized visual encoder (Head (Random)), and a policy head trained on a imagenet pretrained visual encoder (Head (ImageNet)) to show the importance of using learned visual features.

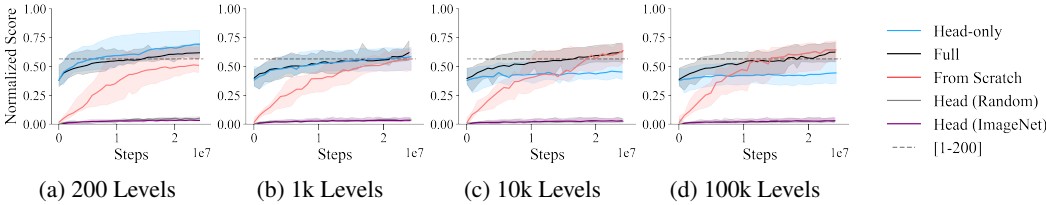

Figure 6: We compare fine-tuning the policy head (Head-only) versus the entire policy network (Full). The policy is trained on 200 training levels ([1 - 200]), and fine-tuned on (a) 200, (b) 1k, (c) 10k, and (d) 100k new test levels. We additionally include the learning curve of a policy trained from scratch (From Scratch), a policy head trained on a randomly initialized visual encoder (Head (Random)), and a policy head trained on a imagenet pretrained visual encoder (Head (ImageNet)) to show the ability for the visual features to transfer. The plots show the normalized reward averaged across all games (see results on individual games in Appendix C).

adapting to new levels $(201 - 400)$. Instead, it is the policy head feature that fails to transfer to new levels and requires further finetuning.

However, the strategy of training a policy on training levels $(1 - 200)$ and then finetuning the policy head on new levels has limitations. As the number of new levels increases, it becomes harder to finetune the policy head on those levels and recover the average policy returns of the training levels. From Figure 6, we can see that finetuning the policy head on 200 and 1k new levels recovers the average policy returns of the training levels. But, it fails to do the same for 10k and 100k new levels.

## 9 DISCUSSION

**How to use visual features?** Most of the work on visual transfer in RL has focused on learning generalizable visual features. However, another related problem setting is policy adaptation to new levels. Section 8 shows that visual features learned by PPO can be kept fixed when adapting to new levels but the policy head, tasked to combine visual features to predict actions, requires finetuning on new levels. This suggests a fruitful avenue for future work: instead of learning better visual features, focus on how to leverage the learned features to solve a new task.

**Towards better data augmentation.** UCB-DrAC performs similar to PPO baseline when averaged across all the games and doesn't fully address the problem of learning invariant features. Specifically, it is significantly worse as compared to training PPO with fixed texture. This indicates that better data augmentation techniques are required to make the policy invariant to irrelevant visual appearances. We show that one way to overcome this challenge is by combining UCB-DrAC with theme randomization. However, theme randomization requires task-specific knowledge of how the theme varies across different variants of the task. In general, such knowledge may not available to the agents. An exciting area of future investigation is to develop data augmentation method that can automatically discover task-relevant augmentations.

**Further investigation of auxiliary losses.** We show PPO, with inverse model regularization, significantly improves upon PPO despite being a simple change. Similarly, iDAAC made use of auxiliary losses to focus on task-relevant features of the environment. This indicates the need for further investigation into auxiliary losses for better policy generalization.

**Towards compute-efficient hyperparameter tuning.** We show that state-of-the-art (SOTA) methods that improve policy generalization heavily rely on careful hyperparameter selection for each game. However, hyperparameter tuning via grid search on each game is compute-intensive. Therefore, automatically tuning hyperparameters in a compute-efficient manner is a fruitful avenue for future works.

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

# A  PPO WITH INVERSE MODEL REGULARIZATION

PPO (Schulman et al., 2017) is an actor-critic algorithm that alternates between sampling data with environment interaction and optimizing objective function with stochastic gradient descent. Let the policy being optimizes be $\pi_\theta$ and the associated value function be $V_{\theta,\psi}$. PPO maximizes

$$J_{\text{PPO}} = J_{\pi_\theta} - \alpha_1 J_V + \alpha_2 S_{\pi_\theta}$$

where $S_{\pi_\theta}$ is the entropy for aiding exploration,

$$S_{\pi_\theta} = \mathbb{E}_{s \in \pi_{\theta_{\text{old}}}} \left[ - \sum_{a \in \mathcal{A}} \pi_\theta(a|s) \log \pi_\theta(a|s) \right]$$

$J_V$ is the value function loss,

$$J_V = \mathbb{E}_{s \in \pi_{\theta_{\text{old}}}} [(V_{\theta,\psi}(s) - V_t^{\text{target}})^2]$$

and $J_{\pi_\theta}$ is the policy objective term

$$J_{\pi_\theta} = \mathbb{E}_{(s,a) \in \pi_{\theta_{\text{old}}}} \left[ \min(r_\theta \hat{A}, \text{clip}(r_\theta, 1 - \epsilon, 1 + \epsilon) \hat{A}) \right]$$

Here, $r_\theta = \frac{\pi_\theta(a|s)}{\pi_{\theta_{\text{old}}}(a|s)}$ is the importance weight for estimating the advantage function and $\hat{A} = \hat{A}_{\theta_{\text{old}}}(s, a)$ is estimated advantage function for $\pi_{\theta_{\text{old}}}$ using returns and value function $V_{\theta,\psi}(s)$.

We make use of inverse models to regularize policy features for improved performance and better generalization. Let $z_t = \pi_{\theta_1}^{enc}(s_t)$. If $f_\phi$ is the inverse model and $(s_t, a_t, s_{t+1})$ is the state action state tuple coming from $\pi_{\theta_{\text{old}}}$, then this regularization minimizes $J_{\text{inv}} = \mathcal{CE}(a_t, f_\phi(z_t, z_{t+1}))$ where $\mathcal{CE}$ is the cross-entropy loss. Overall, PPO with inverse model regularization maximizes

$$J_{\text{PPO}} = J_{\pi_\theta} - \alpha_1 J_V + \alpha_2 S_{\pi_\theta} - \alpha_{\text{inv}} J_{\text{inv}}$$

where $\alpha_{\text{inv}}$ is the regularization constant associated with inverse model regurlarization loss.

# B  HYPERPARAMETERS

We use the default hyperparameters from PPO (Schulman et al., 2017), UCB-DrAC (Raileanu et al., 2020) and DAAC/iDAAC (Raileanu & Fergus, 2021). DrAC-FT and DrAC-TR uses same hyperparameters as UCB-DrAC (DrAC). PPO with inverse model regularization only introduces an extra hyperparameter $\alpha_{\text{inv}}$. We write all the hyperparameters in table 2 and 3. While table 2 describes the best single set of hyperparameters for different methods selected using validation levels $201 - 400$, table 3 details game-specific changes to some of those hyperparameters for improved performance.

# C  PROCGEN RESULTS

Table 4 and 5 describe scores on train levels (1-200) and test levels across all procgen games for PPO, UCB-DrAC, DrAC-FT (UCB-DrAC with fixed theme games), DrAC-TR (UCB-DrAC with randomized theme games), DAAC, iDAAC, and PPO inv-m (PPO with inverse model regularization). Figure 7 places games next to the RL algorithm on which they perform the best. Some games are excluded as they perform the best on two or more RL algorithm. Figures 8, 9, and 10 show the learning curves for finetuning the policy head as well as the full policy on 1k, 10k and 100k new levels respectively. The policy was originally trained on levels 1-200 with PPO. End-to-end training of policy and training of policy head with randomly initialized visual encoder on new levels are also included as baselines. Finally, table 6 and 7 give train and test accuracy of level classifier trained on the encoder features of policies learned with PPO on 200 levels, with PPO on 100k levels, with DAAC on 200 levels, and with iDAAC on 200 levels on all procgen games.

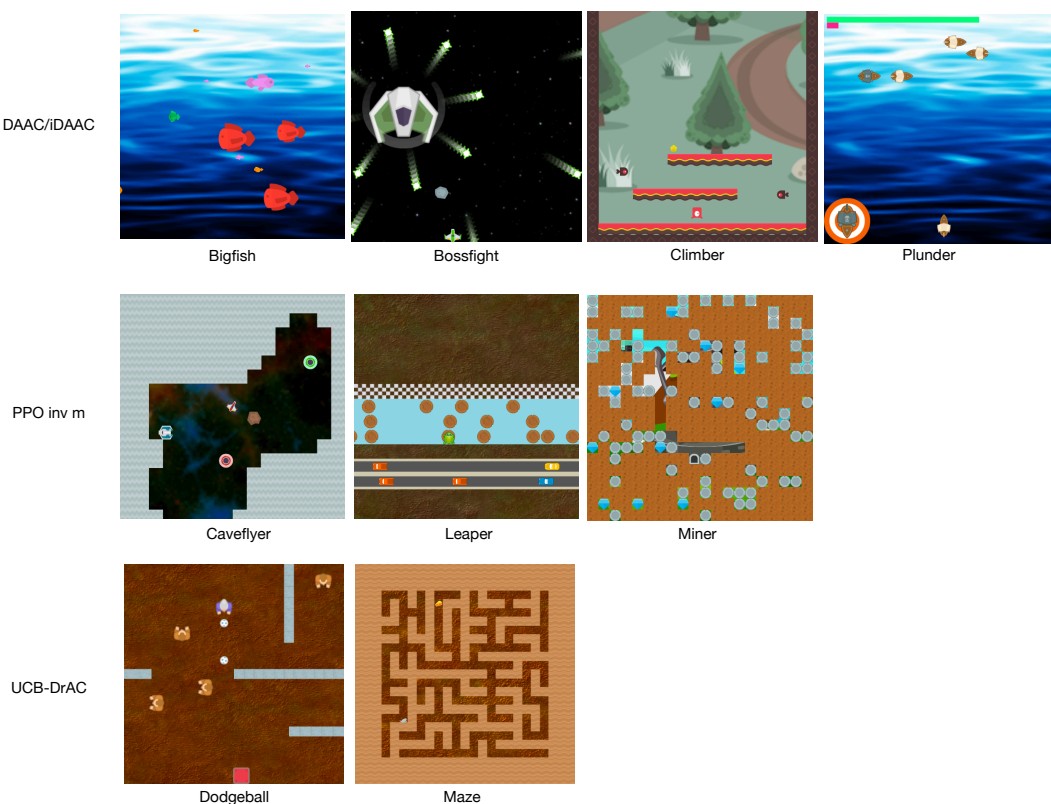

Figure 7: Visualization of games placed next to the RL algorithm on which they perform the best. Some games are excluded as they perform the best on two or more RL algorithms.

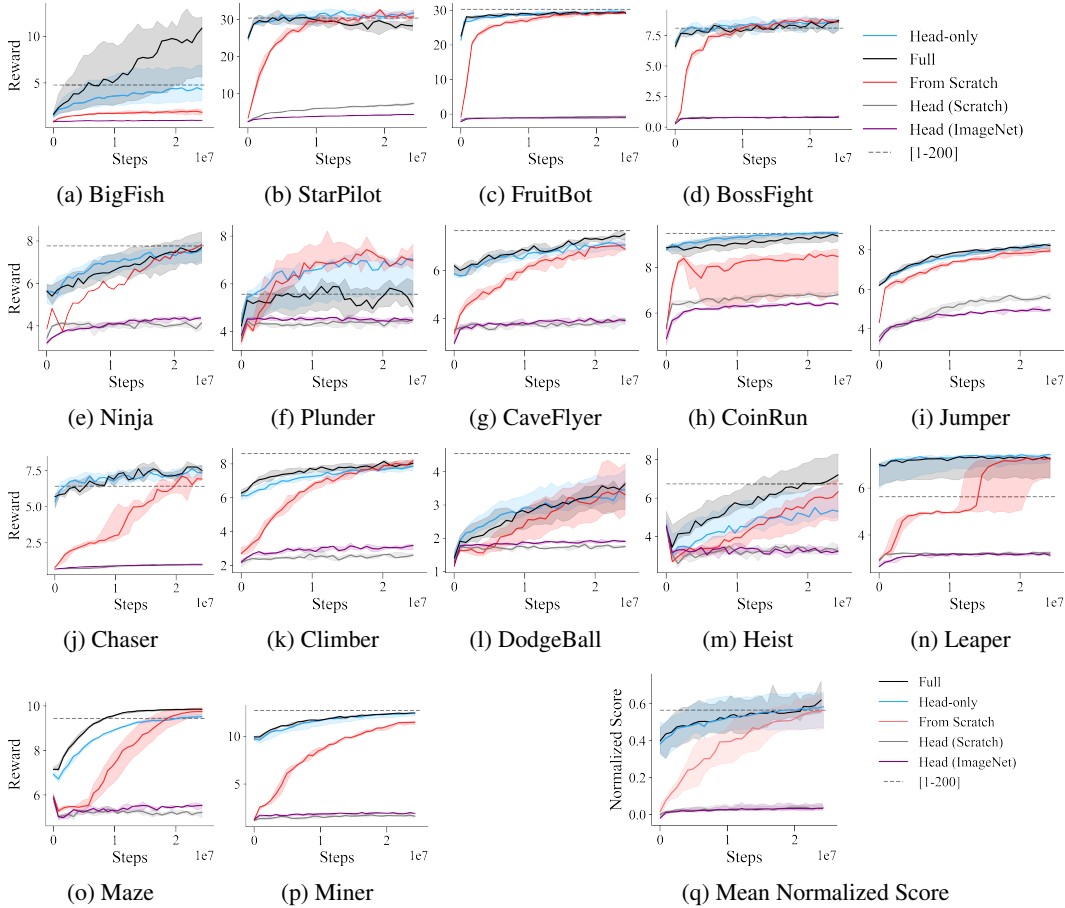

Figure 8: **Transfer**: We compare fine-tuning the policy head (Head-only) versus the entire policy network (Full) on 1k new levels. The policy is trained on 200 levels. Fine tuning just the policy head allows us to recover this performance on the training levels ([1 - 200]). We additionally include the learning curve of a policy trained from scratch (From Scratch), a policy head trained on a randomly initialized visual encoder (Head (Random)), and a policy head trained on a imagenet pretrained visual encoder (Head (ImageNet)) to show the importance of using learned visual features.

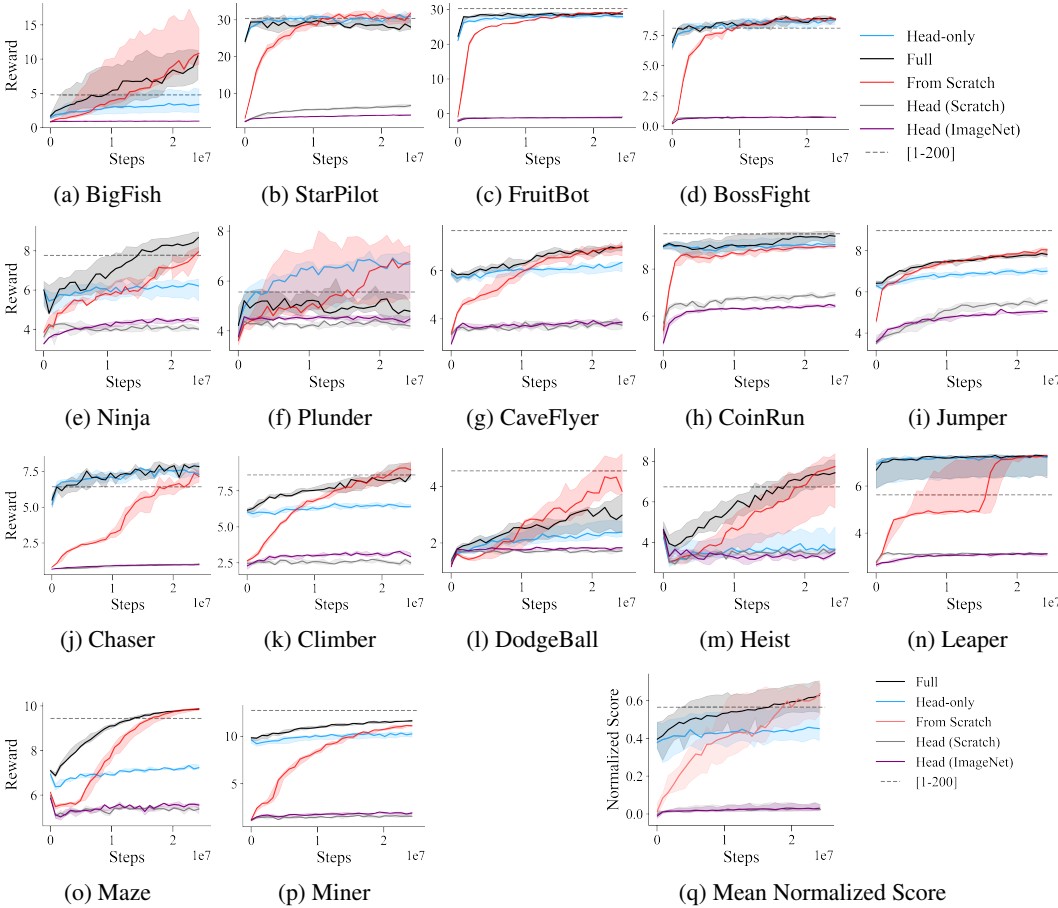

Figure 9: **Transfer**: We compare fine-tuning the policy head (Head-only) versus the entire policy network (Full) on 10k new levels. The policy is trained on 200 levels. Fine tuning only the policy head fails to recover the performance on the training levels. We additionally include the learning curve of a policy trained from scratch (From Scratch), a policy head trained on a randomly initialized visual encoder (Head (Random)), and a policy head trained on a imagenet pretrained visual encoder (Head (ImageNet)) to show the importance of using learned visual features.

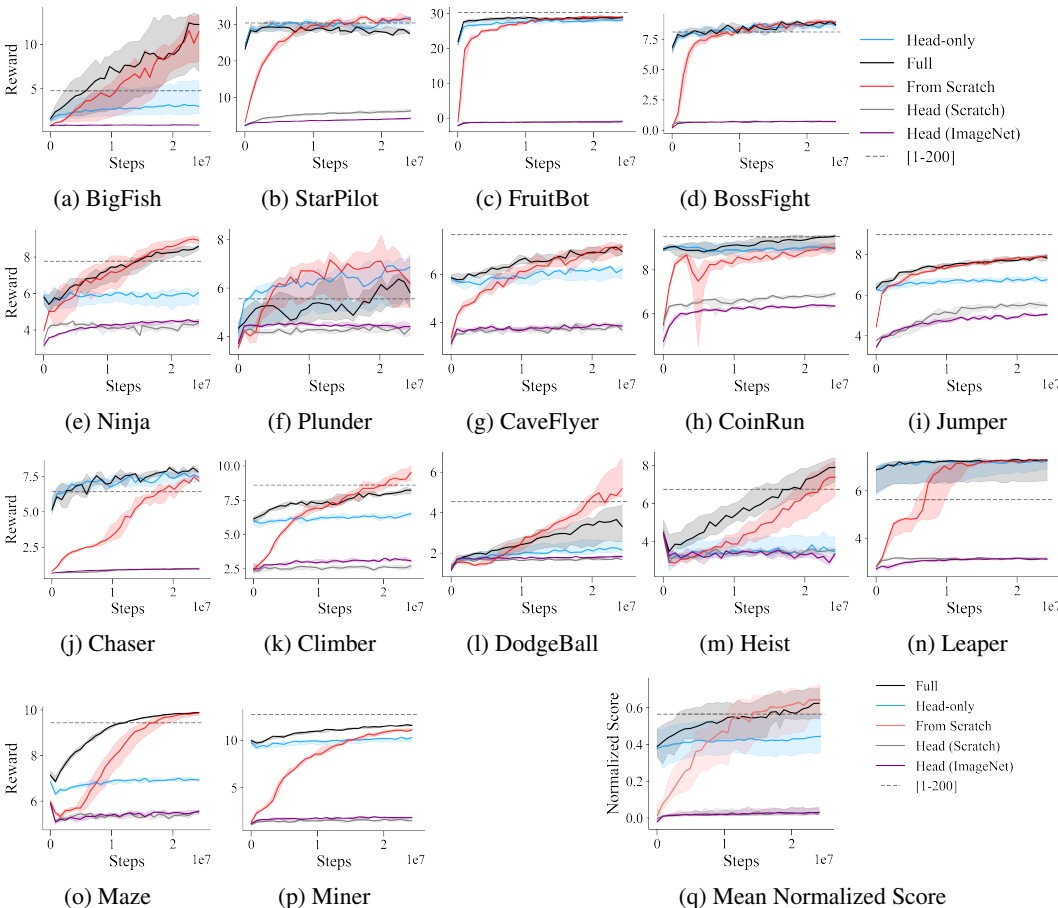

Figure 10: **Transfer**: We compare fine-tuning the policy head (Head-only) versus the entire policy network (Full) on 100k new levels. The policy is trained on 200 levels. Fine tuning only the policy head fails to recover the performance on the training levels. We additionally include the learning curve of a policy trained from scratch (From Scratch), a policy head trained on a randomly initialized visual encoder (Head (Random)), and a policy head trained on a imagenet pretrained visual encoder (Head (ImageNet)) to show the importance of using learned visual features.

Table 2: Best single set of hyperparameters for PPO, UCB-DrAC, DAAC and iDAAC selected using validation levels $201 - 400$.

| Hyperparameter | Value |
|---|---|
| $\gamma$ | 0.999 |
| $\lambda$ | 0.95 |
| # timesteps per rollout | 256 |
| # epochs per rollout | 3 |
| # minibatches per epoch | 8 |
| entropy bonus ($\alpha_2$) | 0.01 |
| value loss coefficient ($\alpha_1$) | 0.01 |
| clip range | 0.2 |
| reward normalization | yes |
| learning rate | 5e-4 |
| # workers | 1 |
| # environments per worker | 64 |
| # total timesteps | 25M |
| optimizer | Adam |
| LSTM | no |
| frame stack | no |
| $\alpha_{\mathrm{inv}}$ | 0.4 |
| UCB augmentation coefficient ($\alpha_r$) | 0.1 |
| UCB exploration coefficient (c) | 0.1 |
| UCB window length (K) | 10 |
| # DAAC/iDAAC actor epochs per rollout | 1 |
| # DAAC/iDAAC value epochs per rollout ($v_e$) | 9 |
| # DAAC/iDAAC value update frequency ($v_f$) | 1 |
| DAAC/iDAAC advantage loss coefficient ($\alpha_a$) | 0.25 |
| iDAAC order loss coefficient ($\alpha_i$) | 0.001 |

Table 3: Game-specific changes to some hyperparameters for improved performance

| Hyperparameter | $v_e$ | $v_f$ | $\alpha_a$ | $\alpha_i$ | $\alpha_{\mathrm{inv}}$ |
|---|---|---|---|---|---|
| plunder | 1 | 8 | 0.3 | 0.1 | 0.4 |
| chaser | 1 | 1 | 0.15 | 0.001 | 0.1 |
| miner | 9 | 32 | 0.25 | 0.1 | 1 |
| climber | 9 | 1 | 0.05 | 0.001 | 0.01 |
| bigfish | 9 | 32 | 0.05 | 0.01 | 0.1 |
| dodgeball | 9 | 32 | 0.25 | 0.001 | 0.1 |
| maze | 9 | 1 | 0.25 | 0.001 | 0.01 |
| leaper | 9 | 1 | 0.25 | 0.001 | 0.4 |
| fruitbot | 9 | 1 | 0.25 | 0.001 | 0.04 |
| bossfight | 9 | 1 | 0.25 | 0.001 | 0.1 |
| jumper | 9 | 1 | 0.25 | 0.001 | 0.1 |
| ninja | 9 | 1 | 0.25 | 0.001 | 0.4 |
| starpilot | 9 | 1 | 0.25 | 0.001 | 0.1 |
| coinrun | 9 | 1 | 0.25 | 0.001 | 0.004 |
| heist | 9 | 1 | 0.25 | 0.001 | 0.4 |
| caveflyer | 9 | 1 | 0.25 | 0.001 | 1 |

Table 4: Procgen scores on train levels (1-200) after training on 25M environment steps. MNS is shorthand for median normalized score across games. The mean and standard deviation are computed using 5 different seeds.

| Game | PPO | UCB-DrAC | DrAC-FT | DrAC-TR | DAAC | iDAAC | PPO inv-m |
|---|---|---|---|---|---|---|---|
| bigfish | $5.3 \pm 1.2$ | $19.3 \pm 1.2$ | $27.2 \pm 1.5$ | $24.2 \pm 2.0$ | $20.1 \pm 1.6$ | $21.8 \pm 1.8$ | $17.3 \pm 6.9$ |
| bossfight | $8.2 \pm 0.4$ | $8.1 \pm 0.8$ | $11.0 \pm 0.7$ | $8.8 \pm 0.5$ | $10.0 \pm 0.4$ | $10.4 \pm 0.4$ | $9.0 \pm 0.6$ |
| caveflyer | $7.8 \pm 0.4$ | $7.2 \pm 0.4$ | $9.2 \pm 1.1$ | $8.2 \pm 1.3$ | $5.8 \pm 0.4$ | $6.2 \pm 0.6$ | $9.4 \pm 0.3$ |
| chaser | $6.2 \pm 1.0$ | $7.1 \pm 1.0$ | $6.9 \pm 0.7$ | $7.6 \pm 0.6$ | $6.9 \pm 1.2$ | $7.5 \pm 0.8$ | $6.6 \pm 0.7$ |
| climber | $8.4 \pm 0.4$ | $8.6 \pm 0.3$ | $11.9 \pm 0.2$ | $9.5 \pm 0.8$ | $10.0 \pm 0.3$ | $10.2 \pm 0.7$ | $8.8 \pm 0.3$ |
| coinrun | $9.5 \pm 0.2$ | $8.4 \pm 1.4$ | $8.0 \pm 1.2$ | $9.3 \pm 0.1$ | $9.8 \pm 0.0$ | $9.8 \pm 0.1$ | $9.7 \pm 0.1$ |
| dodgeball | $4.3 \pm 0.4$ | $9.0 \pm 1.0$ | $12.1 \pm 0.5$ | $10.2 \pm 1.3$ | $5.2 \pm 0.4$ | $4.9 \pm 0.3$ | $8.3 \pm 0.8$ |
| fruitbot | $30.2 \pm 0.5$ | $29.2 \pm 1.1$ | $31.0 \pm 0.5$ | $30.3 \pm 0.6$ | $29.7 \pm 0.4$ | $29.1 \pm 0.7$ | $30.5 \pm 0.4$ |
| heist | $7.0 \pm 1.1$ | $6.9 \pm 0.6$ | $7.0 \pm 0.4$ | $7.1 \pm 1.0$ | $5.2 \pm 0.7$ | $4.5 \pm 0.3$ | $7.1 \pm 0.5$ |
| jumper | $8.9 \pm 0.1$ | $8.1 \pm 0.2$ | $9.2 \pm 0.1$ | $8.8 \pm 0.3$ | $8.6 \pm 0.3$ | $8.7 \pm 0.2$ | $9.1 \pm 0.1$ |
| leaper | $6.1 \pm 1.1$ | $4.1 \pm 2.0$ | $6.1 \pm 1.2$ | $5.0 \pm 2.7$ | $8.0 \pm 1.1$ | $8.3 \pm 0.7$ | $9.1 \pm 1.3$ |
| maze | $9.5 \pm 0.2$ | $8.8 \pm 0.3$ | $9.0 \pm 0.3$ | $9.4 \pm 0.4$ | $6.6 \pm 0.4$ | $6.4 \pm 0.5$ | $9.7 \pm 0.2$ |
| miner | $12.7 \pm 0.1$ | $12.2 \pm 0.4$ | $12.8 \pm 0.0$ | $12.6 \pm 0.1$ | $11.3 \pm 0.9$ | $11.5 \pm 0.5$ | $12.8 \pm 0.1$ |
| ninja | $7.7 \pm 0.3$ | $6.7 \pm 0.6$ | $9.7 \pm 0.1$ | $9.2 \pm 0.1$ | $8.8 \pm 0.2$ | $8.9 \pm 0.3$ | $9.2 \pm 0.3$ |
| plunder | $5.5 \pm 0.8$ | $7.1 \pm 2.0$ | $16.0 \pm 3.3$ | $9.5 \pm 2.8$ | $22.5 \pm 2.8$ | $24.6 \pm 1.6$ | $9.0 \pm 3.8$ |
| starpilot | $30.3 \pm 1.5$ | $32.5 \pm 3.9$ | $40.3 \pm 3.0$ | $34.3 \pm 1.9$ | $38.0 \pm 2.6$ | $38.6 \pm 2.2$ | $36.6 \pm 2.1$ |
| MNS | $0.57 \pm 0.05$ | $0.55 \pm 0.09$ | $0.72 \pm 0.07$ | $0.66 \pm 0.08$ | $0.62 \pm 0.06$ | $0.63 \pm 0.05$ | $0.69 \pm 0.06$ |

Table 5: Procgen scores on test levels after training on 25M environment steps. MNS is shorthand for median normalized score across games. The mean and standard deviation are computed using 3 different seeds.

| Game | PPO | UCB-DrAC | DrAC-FT | DrAC-TR | DAAC | iDAAC | PPO inv-m |
|---|---|---|---|---|---|---|---|
| bigfish | $1.7 \pm 0.3$ | $15.1 \pm 3.8$ | $26.2 \pm 2.2$ | $20.9 \pm 2.7$ | $17.8 \pm 1.4$ | $18.5 \pm 1.2$ | $12.3 \pm 6.2$ |
| bossfight | $7.8 \pm 1.1$ | $7.3 \pm 0.2$ | $11.2 \pm 0.6$ | $9.7 \pm 0.8$ | $9.6 \pm 0.5$ | $9.8 \pm 0.5$ | $8.6 \pm 1.1$ |
| caveflyer | $6.0 \pm 0.7$ | $5.0 \pm 0.1$ | $7.0 \pm 1.5$ | $6.9 \pm 1.2$ | $4.6 \pm 0.2$ | $5.0 \pm 0.2$ | $7.0 \pm 0.2$ |
| chaser | $5.7 \pm 1.0$ | $7.1 \pm 0.3$ | $6.4 \pm 0.8$ | $6.9 \pm 1.2$ | $6.6 \pm 1.2$ | $6.8 \pm 1.2$ | $2.7 \pm 0.2$ |
| climber | $5.7 \pm 0.7$ | $6.9 \pm 0.4$ | $11.2 \pm 0.2$ | $8.2 \pm 1.1$ | $7.8 \pm 0.2$ | $8.3 \pm 0.2$ | $5.9 \pm 1.2$ |
| coinrun | $8.9 \pm 0.2$ | $7.6 \pm 0.4$ | $7.3 \pm 1.4$ | $8.4 \pm 0.3$ | $9.2 \pm 0.2$ | $9.4 \pm 0.2$ | $9.1 \pm 0.1$ |
| dodgeball | $1.5 \pm 0.4$ | $6.0 \pm 0.7$ | $9.4 \pm 0.2$ | $7.7 \pm 0.9$ | $3.3 \pm 0.5$ | $3.2 \pm 0.5$ | $3.8 \pm 0.8$ |
| fruitbot | $25.1 \pm 1.0$ | $25.6 \pm 0.2$ | $27.8 \pm 1.3$ | $27.9 \pm 1.7$ | $28.6 \pm 0.6$ | $27.9 \pm 0.6$ | $27.0 \pm 1.1$ |
| heist | $2.7 \pm 0.4$ | $3.2 \pm 1.1$ | $3.6 \pm 2.3$ | $3.1 \pm 1.4$ | $3.3 \pm 0.2$ | $3.5 \pm 0.2$ | $3.6 \pm 0.5$ |
| jumper | $5.9 \pm 0.4$ | $5.5 \pm 0.3$ | $7.7 \pm 0.5$ | $7.1 \pm 0.4$ | $6.5 \pm 0.4$ | $6.3 \pm 0.4$ | $6.3 \pm 0.3$ |
| leaper | $6.4 \pm 1.1$ | $4.0 \pm 1.6$ | $6.0 \pm 1.7$ | $5.5 \pm 2.4$ | $7.3 \pm 1.1$ | $7.7 \pm 1.1$ | $8.7 \pm 0.6$ |
| maze | $6.2 \pm 0.5$ | $6.9 \pm 0.5$ | $8.1 \pm 0.2$ | $7.1 \pm 1.3$ | $5.5 \pm 0.2$ | $5.6 \pm 0.2$ | $6.0 \pm 0.7$ |
| miner | $9.0 \pm 0.7$ | $9.1 \pm 0.3$ | $11.2 \pm 0.4$ | $10.0 \pm 0.9$ | $8.6 \pm 0.9$ | $9.5 \pm 0.9$ | $10.1 \pm 0.4$ |
| ninja | $6.2 \pm 0.5$ | $5.0 \pm 0.3$ | $9.3 \pm 0.3$ | $8.6 \pm 0.4$ | $6.8 \pm 0.4$ | $6.8 \pm 0.4$ | $6.9 \pm 0.4$ |
| plunder | $4.7 \pm 0.5$ | $6.0 \pm 0.9$ | $13.6 \pm 2.1$ | $8.6 \pm 2.0$ | $20.7 \pm 3.3$ | $23.3 \pm 3.3$ | $8.3 \pm 3.2$ |
| starpilot | $28.8 \pm 3.8$ | $31.5 \pm 4.3$ | $42.1 \pm 3.5$ | $35.0 \pm 1.6$ | $36.4 \pm 2.8$ | $37.0 \pm 2.3$ | $36.7 \pm 3.0$ |
| MNS | $0.36 \pm 0.06$ | $0.37 \pm 0.1$ | $0.59 \pm 0.10$ | $0.51 \pm 0.11$ | $0.48 \pm 0.06$ | $0.50 \pm 0.06$ | $0.47 \pm 0.07$ |

Table 6: Train accuracy of level classifier trained on the encoder features of policies learned with PPO on 200 levels, with PPO on 100k levels, with DAAC on 200 levels, and with iDAAC on 200 levels on all procgen games.

|  | PPO (200) | PPO (100k) | DAAC | iDAAC |
|---|---|---|---|---|
| bigfish | $1.00 \pm 0.01$ | $1.00 \pm 0.00$ | $1.00 \pm 0.00$ | $1.00 \pm 0.00$ |
| bossfight | $1.00 \pm 0.01$ | $0.99 \pm 0.01$ | $1.00 \pm 0.00$ | $1.00 \pm 0.00$ |
| caveflyer | $1.00 \pm 0.00$ | $1.00 \pm 0.00$ | $0.98 \pm 0.03$ | $1.00 \pm 0.00$ |
| chaser | $1.00 \pm 0.00$ | $1.00 \pm 0.00$ | $1.00 \pm 0.00$ | $1.00 \pm 0.00$ |
| climber | $0.97 \pm 0.02$ | $0.95 \pm 0.01$ | $0.97 \pm 0.00$ | $0.97 \pm 0.01$ |
| coinrun | $0.67 \pm 0.14$ | $0.66 \pm 0.27$ | $1.00 \pm 0.00$ | $0.98 \pm 0.02$ |
| dodgeball | $1.00 \pm 0.00$ | $1.00 \pm 0.00$ | $1.00 \pm 0.00$ | $1.00 \pm 0.00$ |
| fruitbot | $0.97 \pm 0.00$ | $0.90 \pm 0.01$ | $0.70 \pm 0.07$ | $0.66 \pm 0.03$ |
| heist | $1.00 \pm 0.00$ | $1.00 \pm 0.00$ | $1.00 \pm 0.00$ | $1.00 \pm 0.00$ |
| jumper | $1.00 \pm 0.00$ | $1.00 \pm 0.00$ | $1.00 \pm 0.00$ | $0.99 \pm 0.01$ |
| leaper | $1.00 \pm 0.00$ | $1.00 \pm 0.00$ | $1.00 \pm 0.00$ | $1.00 \pm 0.00$ |
| maze | $0.99 \pm 0.01$ | $1.00 \pm 0.00$ | $1.00 \pm 0.00$ | $1.00 \pm 0.00$ |
| miner | $0.92 \pm 0.03$ | $1.00 \pm 0.00$ | $0.82 \pm 0.02$ | $0.78 \pm 0.03$ |
| ninja | $0.75 \pm 0.02$ | $0.83 \pm 0.04$ | $0.74 \pm 0.07$ | $0.73 \pm 0.03$ |
| plunder | $1.00 \pm 0.00$ | $1.00 \pm 0.00$ | $0.99 \pm 0.01$ | $0.99 \pm 0.00$ |
| starpilot | $1.00 \pm 0.00$ | $1.00 \pm 0.00$ | $1.00 \pm 0.00$ | $1.00 \pm 0.00$ |

Table 7: Test accuracy of level classifier trained on the encoder features of policies learned with PPO on 200 levels, with PPO on 100k levels, with DAAC on 200 levels, and with iDAAC on 200 levels on all procgen games.

|  | PPO (200) | PPO (100k) | DAAC | iDAAC |
|---|---|---|---|---|
| bigfish | $0.94 \pm 0.04$ | $0.74 \pm 0.12$ | $0.97 \pm 0.01$ | $0.97 \pm 0.01$ |
| bossfight | $0.78 \pm 0.03$ | $0.69 \pm 0.02$ | $0.78 \pm 0.04$ | $0.76 \pm 0.02$ |
| caveflyer | $1.00 \pm 0.00$ | $0.94 \pm 0.02$ | $0.82 \pm 0.03$ | $0.80 \pm 0.03$ |
| chaser | $1.00 \pm 0.00$ | $0.96 \pm 0.01$ | $0.89 \pm 0.03$ | $0.85 \pm 0.02$ |
| climber | $0.93 \pm 0.02$ | $0.72 \pm 0.02$ | $0.85 \pm 0.05$ | $0.79 \pm 0.03$ |
| coinrun | $0.38 \pm 0.10$ | $0.35 \pm 0.22$ | $0.95 \pm 0.03$ | $0.92 \pm 0.03$ |
| dodgeball | $1.00 \pm 0.00$ | $1.00 \pm 0.00$ | $1.00 \pm 0.00$ | $0.99 \pm 0.01$ |
| fruitbot | $0.92 \pm 0.01$ | $0.79 \pm 0.02$ | $0.57 \pm 0.06$ | $0.54 \pm 0.03$ |
| heist | $1.00 \pm 0.00$ | $1.00 \pm 0.00$ | $0.99 \pm 0.00$ | $0.99 \pm 0.00$ |
| jumper | $1.00 \pm 0.00$ | $0.99 \pm 0.00$ | $0.88 \pm 0.04$ | $0.89 \pm 0.04$ |
| leaper | $1.00 \pm 0.00$ | $1.00 \pm 0.00$ | $0.99 \pm 0.01$ | $0.98 \pm 0.00$ |
| maze | $0.96 \pm 0.02$ | $0.99 \pm 0.00$ | $1.00 \pm 0.00$ | $1.00 \pm 0.00$ |
| miner | $0.92 \pm 0.03$ | $1.00 \pm 0.00$ | $0.81 \pm 0.01$ | $0.78 \pm 0.03$ |
| ninja | $0.70 \pm 0.04$ | $0.74 \pm 0.03$ | $0.71 \pm 0.06$ | $0.69 \pm 0.03$ |
| plunder | $1.00 \pm 0.00$ | $1.00 \pm 0.00$ | $0.99 \pm 0.02$ | $0.99 \pm 0.00$ |
| starpilot | $1.00 \pm 0.00$ | $1.00 \pm 0.00$ | $1.00 \pm 0.00$ | $1.00 \pm 0.00$ |

