# OpenReview forum: "Understanding the Generalization Gap in Visual Reinforcement Learning"
_ICLR.cc/2022/Conference — ICLR 2022 Submitted_

### Official Review · Reviewer_SFef · 2021-10-19

**Correctness:** 2
**Technical Novelty And Significance:** 1
**Empirical Novelty And Significance:** 1
**Recommendation:** 3
**Confidence:** 4

**Main Review:**

**Strengths**

The paper attempts to address why policies learned for video games have generalization issues, which is an important problem and helps us design better models and learning algorithms in the future.

**Weakness**

The main issue is that there is almost no novel finding in the paper. Most of the conclusions are already known. See details below:

- Section 4: I am wondering why the policy should not have level-specific features when it is trained for it. Also, the experiment is not flawless. Two states in train and test might be very close to each other. This doesn't mean that the policy has level-specific features.

- Section 5: The paper shows an inverse model (inferring the action between two states) achieves good performance. That is also already known, and several others papers have shown that. Some of these papers have been cited in the paper as well.

- Section 6: The conclusion of this section is that the performance drops if a single set of parameters is used for all games, which is expected.

- Section 7:  It is mentioned that "we see that end-to-end training of policy on levels 201 − 400 is less sample-efficient than finetuning the policy head as well as finetuning the entire policy". This statement is not correct. The blue and black curves in Figure 5 are almost identical or have a large overlap for most games.

- Some of the conclusions are not aligned with the observations of other works. For example, the paper recommends that "This suggests a fruitful avenue for future work: instead of learning better visual features, focus on how to leverage the learned features to solve a new task". However, works such as [A] show better visual encoders are important for generalization to unseen environments.

**Comments about the details:**

- It is mentioned that "Figure 5 shows that the average returns of the policy on levels 201−400 become similar to the average
returns of the policy on levels 1 − 200". I am wondering if average returns are comparable. These are two different sets of layouts.

- It is mentioned that  "Furthermore, we implemented a variant of the ProcGen suite that uses a separate randomization
process for the theme versus the game layouts". I am not sure what this means. It is unclear how the same randomization process was used for both theme and layout in the other scenario.

[A] Wijmans et al., How to Train PointGoal Navigation Agents on a (Sample and Compute) Budget, 2020.

**Summary Of The Paper:**

The paper studies generalization of policies trained for video games. More specifically, it analyzes transfer to new game situations via changing the theme (appearance) or the level (layout) of the game. The Procgen framework is used to procedurally generate new levels or themes to evaluate a set of hypotheses. The paper investigates (1) data augmentation, (2) policy generalization via domain confusion (3) generalization via auxiliary tasks (4) effects of hyperparameter selection and (5) adaptation to new levels.

**Summary Of The Review:**

First, I do not see any novel conclusion in the paper:

(1) It is already shown by several papers that inverse models are helpful.

(2) It is already known that data augmentation techniques are required for better transfer.

(3) It is expected that using the same parameters for all games results in inferior performance compared to using game-specific parameters.

Second, the conclusion that improving visual encoders is not that important contradicts other works.

---

> ### Author Response · Authors · 2021-11-23
> **Author's reply (1/2)**
>
> We would like to thank the reviewer for their constructive feedback. We address specific point below:
>
> *First, I do not see any novel conclusion in the paper:*
>
> *(1) It is already shown by several papers that inverse models are helpful.*
>
> *(2) It is already known that data augmentation techniques are required for better transfer.*
>
> *(3) It is expected that using the same parameters for all games results in inferior performance compared to using game-specific parameters.*
>
> **RESPONSE:**
>
> (1) While past works have shown that inverse models improve the training performance, its role in improving generalization of policies is underappreciated. Infact, in a recent work by Ye et al. (CVPR 2021), the authors explicitly said
>
> > While auxiliary tasks are commonly used to improve agent performance [13,23], we believe their role in preventing overfitting to complex environments is underappreciated.
> >
>
> In this work, we find that inverse model regularization (an auxiliary task) with PPO is competent with SOTA data augmentation methods such as UCB-DrAC. Our work provides a comprehensive comparison on 16 different Procgen games. Ye et al (2021) make a similar observation on the Habitat domain. While tasks in Habitat consist of navigating an unseen environment (from a first-person perspective) to find a particular object, Procgen consists of platformer and arcade-style games (from a third-person perspective) where we need to solve an unseen level during test time. Therefore, our work and Ye et al. combined show that using auxiliary tasks can be used to improve policy generalization on many different domains.
>
> (2) Indeed, better data augmentation will improve policy generalization. However, this is not what we intended to communicate and we are sorry for the lack of clarity. It is widely believed in the community that data augmentation (i.e., UCB-DrAC) can make the learned policy invariant to appearance changes. In contrast, what we show is that the SOTA data augmentation method (UCB-DrAC) is insufficient to close the generalization gap caused by such changes (i.e., theme variation). Our results show that there is a substantial performance gap between UCB-DrAC and a theme-invariant baseline that only needs to generalize over layouts. Our key insight, therefore, is that **there is substantial room for improvement over current data augmentation methods,** which we don't believe is trivial.
>
> (3) While it is expected that using game-specific hyper-parameters leads to improved performance, it does raise the question if the improvement in performance of new methods comes from careful hyper-parameter tuning or from the method itself. The baseline PPO uses same hyperparameters across all games.

---

> > ### Author Response · Authors · 2021-11-23
> > **Author's reply (2/2)**
> >
> > *Section 7: It is mentioned that "we see that end-to-end training of policy on levels 201 − 400 is less sample-efficient than finetuning the policy head as well as finetuning the entire policy". This statement is not correct. The blue and black curves in Figure 5 are almost identical or have a large overlap for most games.*
> >
> > **RESPONSE:** The highlighted sentence is a bit misleading, and we are sorry for the confusion. We have corrected it in the updated draft. The sentence should be "*we see that end-to-end training of policy on levels 201 − 400 from scratch is less sample-efficient than finetuning the policy head as well as finetuning the entire policy".* As seen from Figure 5, blue/black figures (finetuning the policy head/finetuning the full policy) reach their final performance faster than the pink curve (training end to end from scratch). Thus, the policy features learned by training it on levels 201-400 is useful and leads to faster transfer to levels 201-400. Since the blue and black curves match, this implies we can keep the 15-layer visual encoder frozen and only finetune the last two fully connected layers when transferring to new levels (i.e., levels 201-400).
> >
> > *Some of the conclusions are not aligned with the observations of other works. For example, the paper recommends that "This suggests a fruitful avenue for future work: instead of learning better visual features, focus on how to leverage the learned features to solve a new task". However, works such as [A] show better visual encoders are important for generalization to unseen environments.*
> >
> > **RESPONSE:** Indeed, good visual encoders are important for transfer to new environments. In fact, Figure 5 shows that training policy head on fixed randomly initialized visual encoder or fixed imagenet pretrained visual encoder leads to poor performance on levels 201-400. However, we argue that the visual encoder obtained by training a policy with PPO on levels 1-200 is sufficient to transfer to new test levels (i.e. levels 201-400) as long as one fine-tunes the policy head. Given most of the current work in RL generalization/transfer is focused on learning better visual encoders, we wanted to draw attention to an alternate problem setting of few-shot adaptation to new levels and show that visual features (learned on train levels i.e. levels 1-200) can be kept frozen in that setting.
> >
> > *It is mentioned that "Figure 5 shows that the average returns of the policy on levels 201−400 become similar to the average returns of the policy on levels 1 − 200". I am wondering if average returns are comparable. These are two different sets of layouts.*
> >
> > **RESPONSE:** We also train a policy from scratch on levels 201-400 (Figure 5, pink curve) and see that its average returns on levels 201-400 is similar to average returns on levels 1-200 of a policy trained on levels 1-200.
> >
> > *It is mentioned that "Furthermore, we implemented a variant of the ProcGen suite that uses a separate randomization process for the theme versus the game layouts". I am not sure what this means. It is unclear how the same randomization process was used for both theme and layout in the other scenario.*
> >
> > **RESPONSE:** In procgen, a level is determined by a random seed (i.e. level seed) which is taken by the generation process as an input. Hence, a level has a fixed theme and a fixed layout. To test theme randomization in procgen, we implemented a variant where a level uses level seed for only layout generation and samples theme randomly at beginning of each episode. In that way, a level has a fixed layout but randomly samples a theme at start of each episode. We have updated the draft to make the language less confusing.
> >
> > **REFERENCES**
> >
> > [1] Joel Ye, Dhruv Batra, Abhishek Das and Eric Wijman. "Auxiliary Tasks and Exploration Enable ObjectNav" CVPR (2021).

---

> > > ### Comment · Reviewer_SFef · 2021-11-28
> > > **Keeping the initial score**
> > >
> > > I checked the rebuttal carefully. There are several unconvincing points in the rebuttal. Here are two examples:
> > > (1) The rebuttal states that using inverse models is under-appreciated. Inverse models have been used extensively in different contexts. So they are not under-appreciated.
> > > (2) The rebuttal states that "It is widely believed in the community that data augmentation can make the learned policy invariant to appearance changes". There is no such belief in the community. Data augmentation improves robustness, but it does not close the gap.
> > >
> > > There are several other issues in the paper. For example, the rebuttal does not answer my concern regarding Section 4.
> > >
> > > Due to these issues, I retain the reject score.

---

### Official Review · Reviewer_eZnU · 2021-11-02

**Correctness:** 2
**Technical Novelty And Significance:** 2
**Empirical Novelty And Significance:** 2
**Recommendation:** 6
**Confidence:** 3

**Main Review:**

The paper asks important questions about generalization theme and task wise and answers them but I am not sure (not saying that they are not, just cannot wrap my head around it) if generalization of the conclusions is justified. Especially considering results showing that hyperparameters must be finetuned for every game separately.

- In abstract starting from "Contrary to..." it's not clear what you meant. Introduction part which states the same info is clear.
- Introduction is sturdy, clean and clear.
- Probably would drop "investigating" in headers and separate motivation/setup/results/discussion parts  of each subsection in more clear way
- Some figures could use better comments (e.g. Figure 2 is ok while Figure 3 lacks similar details like what is the dotted line).
- Figure 2: isn't it an overstatement to say that TR matched FT? on 8-12 out of 15 it look like significantly or slightly  worse.
- The paper is not fully explicit in terms of experimental part but it's usually "like in the <link provided>" so I assume that the sources are explicit enough. Similar approach is taken to descriptions of algos and such - no elaboration and just reference (usually I feel that there is too much but here is too little perhaps). It surely benefits the brevity and cleanness but requires more jumps in other papers than usually if not acquainted with previous work. However I'm not sure whether to count it as positive or negative.



**Summary Of The Paper:**

The paper studies generalization of several previously published visual RL SOTA algorithms with PPO as a baseline in ProcGen environment (2d platformed style, procedurally generated maps). Generalization is studied in terms of theme (colors and styles of objects and background) and tasks (levels with different layouts and specifics things to do). DrAC is shown to not be theme-invariant by comparing result from vanilla version, a version where themes are fixed and version with randomized themes (vanilla version performs worse). For PPO, DAAC and iDAAC, linear classifiers on top of visual feature extractor layers (pretrained on the rl task) are trained to classify states (images from level) based on the level they belong to. Authors claim that accuracy around 90% hints that tested policies contain level specific information regardless of any generalization capabilities. Moreover authors propose a simple auxiliary task "inverse model" for ppo and show that it's comparable with UCB-DRAC. Finally it is mentioned that all methods need careful on per-game basis hyperparameter tuning which requires further research

**Summary Of The Review:**

The whole paper is not groundbreaking but makes its point in a clear and understandable way and seems to be natural consequence of previous research.

---

> ### Author Response · Authors · 2021-11-23
> **Author's reply**
>
> We would like to thank the reviewer for their constructive feedback. We address specific point below:
>
> *In abstract starting from "Contrary to..." it's not clear what you meant. Introduction part which states the same info is clear.*
>
> **RESPONSE:** We have updated the text to improve the abstract.
>
> *Some figures could use better comments (e.g. Figure 2 is ok while Figure 3 lacks similar details like what is the dotted line).*
>
> **RESPONSE:** We have improved the text of Figure 3.
>
> *Figure 2: isn't it an overstatement to say that TR matched FT? on 8-12 out of 15 it look like significantly or slightly worse.*
>
> **RESPONSE:** We intended to say that TR gets much closer to FT when compared to UCB-DrAC. We have updated our text to reflect that.

---

### Official Review · Reviewer_185P · 2021-11-02

**Correctness:** 2
**Technical Novelty And Significance:** 1
**Empirical Novelty And Significance:** 2
**Recommendation:** 3
**Confidence:** 4

**Main Review:**

### Strengths

The paper's overarching goal is to understand why current RL algorithms (aimed at improving generalization) still fall short in certain settings. This is an important problem and answers to this question would be valuable for the community. The paper contains some interesting insights regarding some recently proposed algorithms.

I also liked the findings related to the fact that simple models e.g. using a simple inverse model as auxiliary task can be quite effective at improving generalization in RL and even comparable with other SOTA (aiming to learn domain invariant features) methods like DrAC or DAAC. This suggests a fruitful direction for future work might be understanding which types of auxiliary losses or self-supervised training may lead to good generalization. I think these insights would be valuable for the community and those experiments seemed reasonably well designed for the conclusions to hold.

### Weaknesses

However, the paper does not seem to provide a clear and definite answer to the question. It seems to contain a number of disconnected experiments that answer slightly different questions than the main question.

In addition, some of the claims do not seem to be supported by the experiments. Some of the experiments are lacking important details and are not very well designed to answer the questions they are supposed to.

The paper seems to look at both the setting of zero-shot generalization and the setting of few-shot / fast adaptation to new tasks. However, the authors should clarify which setting they are referring to and use more specific nomenclature throughout the paper since the writing can be confusing at the moment (especially at the beginning of the paper when readers may not know you are investigating multiple settings). In addition, referring to a level as a separate task can be misleading since that typically refers to a different reward and / or transition function, which is not typically the case in Procgen where the levels differ in layout / theme but the overall goal is the same.

In many places throughout the paper, including the figures, it is not clear whether you are referring to the training or test set performance, which makes it difficult to assess the results and conclusions.

1. I am not convinced that the best way to test whether a method is theme invariant is to see if its test performance is the same when training it on levels with the same theme because this may confound better optimization with better generalization (train and test performance)? A better way to test whether a method is theme invariant is to test it on levels that have the same layout but different themes and see if the performance differs from the training levels. However, this does not seem to be the case here, but I found the description to be quite unclear. For DrAC-FT, do you use the exact same layout for the training levels as those used to train DrAC, with the only difference being that the theme is the same for all levels?

2. Does DrAC-TR use observations with same layout but different themes as augmentations or does it simply add more levels to the training set and uses the same types of augmentations as DrAC? Again, I found this section to be missing important details and clarifications. The comparison between DrAC-TR and DrAC isn’t fair since DrAC-TR essentially gets to train on many more levels of a game and because it sees the exact same levels in different themes it has the opportunity to learn theme invariance (perhaps even without the need to explicitly regularize the policy and value function).

3. It would be interesting to compare with PPO-TR to see how it compares to DrAC. You can also apply DrAC but with augmentations given by different themes of the game. An interesting question is whether you need to explicitly regularize the policy and value when you have this type of data and perhaps how many themes do you need to capture this invariance (to any other theme)?

4. In Figure 2, the aggregated results don't seem to differ very much. However, there are games where the differences are significant, and others where they are not. I don't think the text reflects this very well and some of the claims can be misleading.

5. Can you provide some intuition on why DrAC-FT would be better than DrAC? Are they being trained and tested on the exact same dataset?

6. While your experiments seem to suggest that the methods still capture level specific features, while having good generalization, and thus those features are needed to learn generalizable policies, I am not sure the claim is correct. There are level specific features which you need in order to take reasonable actions and others which you do not (so you want to be invariant to these). Perhaps a better test is whether the theme can be predicted (instead of the level), which is typically not useful for solving the task. In contrast, the layout typically is needed to both solve the task and predict the level of the game. It also depends how different the test set is from the train set. If you collected this using a random policy you might end up with essentially identical observations in the train and test set and thus it will be very easy to predict the level (due to memorization). I suggest collected the dataset using a pretrained policy or checking that the test set is completely different from the train set.

The problem might still be that the visual features, while containing the necessary elements to make good decisions, they also contain elements that are superfluous, level-specific, and thus making the policy overfit and fail on certain levels at test time (similar to spurious correlations in SL). I encourage the authors to design experiments aimed at teasing out the two to better understand whether those features are beneficial or not (and in what settings).

7. I think Procgen may not be the best benchmark for this study since it can be difficult to control all the elements of variation across the different levels of a task. I would suggest designing simpler experiments in domains where you can change one variable at a time (e.g. the theme or layout, or agent-goal distance etc.). Perhaps you can then use Procgen to see if the conclusions hold in more complex / different domains.

### Minor
1. Sometimes you refer to UCB-DrAC as UCB-DrAC while other times as DrAC. I think it's better to decide on one of these and mention the convention at the beginning of the paper and then use consistent notation throughout the draft.
2. The related work section should be moved earlier in the paper and definitely before the discussion and conclusion


**Summary Of The Paper:**

This paper looks into why some of the recent methods for improving generalization in deep RL still do not perfectly generalize to new levels of a game. For data augmentation methods, they show that better augmentations which make use of the underlying process that generates the data / ground truth task variation. For approaches that learn domain invariant features, the authors claim that you actually need domain-dependent features to generalize well and that these methods may work well for different reasons (i.e. the additional signal and regularization from the auxiliary losses). They investigate approaches that use data augmentation and domain invariant features and provide some new insights. The authors also show that simple auxiliary tasks can improve generalization policies and that other approaches.

**Summary Of The Review:**

Overall, I think this paper is the beginning of some interesting investigations. However, I am not convinced the experiments described in this paper support the claims made by the authors. In addition, I think it would be better to focus on a one or two questions, clearly state them, and try to answer them as clearly as possible and provide as much insight and depth as possible. In its current form, it is not clear what questions the authors are trying to answer and what the conclusions are.

Given all the above, I do not think the paper is ready for publication. However, I do think the subject of the paper is important and would be of interest for the community. I encourage the authors to take the feedback into account in order to improve the paper, and resubmit at a future conference.

---

> ### Author Response · Authors · 2021-11-23
> **Author's reply (1/3)**
>
> We would like to thank the reviewer for their constructive feedback. We address specific point below:
>
> *The paper seems to look at both the setting of zero-shot generalization and the setting of few-shot / fast adaptation to new tasks. However, the authors should clarify which setting they are referring to and use more specific nomenclature throughout the paper since the writing can be confusing at the moment (especially at the beginning of the paper when readers may not know you are investigating multiple settings). In addition, referring to a level as a separate task can be misleading since that typically refers to a different reward and / or transition function, which is not typically the case in Procgen where the levels differ in layout / theme but the overall goal is the same.*
>
> *In many places throughout the paper, including the figures, it is not clear whether you are referring to the training or test set performance, which makes it difficult to assess the results and conclusions.*
>
> **RESPONSE:** We are sorry for the lack of clarity in writing. We have tried to address these concerns in the updated draft. Rather using the term *performance*, we use train performance or test performance (depending on the context) to avoid confusion. Furthermore, we don't refer to new level as a new task but rather same task in a new env.
>
> *I am not convinced that the best way to test whether a method is theme invariant is to see if its test performance is the same when training it on levels with the same theme because this may confound better optimization with better generalization (train and test performance)?*
>
> **RESPONSE:** It is true that DrAC-FT, which is trained and test on levels with same theme (i.e. train and test levels differ only in layouts), has an easier optimization problem to solve. However, if UCB-DrAC were to obtain theme invariance, it's final test performance would be similar to that of DrAC-FT (which is theme invariant by design). Our results show that there is a substantial test performance gap between UCB-DrAC and DrAC-FT indicating that UCB-DrAC doesn't achieve theme invariance in learned policies. This implies that there is substantial room for improvement over current data augmentation methods.
>
> *A better way to test whether a method is theme invariant is to test it on levels that have the same layout but different themes and see if the performance differs from the training levels. However, this does not seem to be the case here, but I found the description to be quite unclear.*
>
> **RESPONSE:** We are assuming that the reviewer's suggestion is to let train and test levels have the same layout and only differ in themes. However, given procgen is deterministic, training (and testing) on a single layout would lead to policy memorizing an optimal sequence of the actions (using some spurious features) rather than learning theme invariant features. This phenomena of policy memorizing an optimal sequence of the actions in deterministic environments was also observed in Dubey at al. (2018).
>
> *For DrAC-FT, do you use the exact same layout for the training levels as those used to train DrAC, with the only difference being that the theme is the same for all levels?*
>
> **RESPONSE:** Yes, DrAC-FT uses the exact same layout for the training levels as those used to train DrAC, with the only difference being that the theme is the same for all levels.

---

> > ### Author Response · Authors · 2021-11-23
> > **Author's reply (2/3)**
> >
> > *Does DrAC-TR use observations with same layout but different themes as augmentations or does it simply add more levels to the training set and uses the same types of augmentations as DrAC? Again, I found this section to be missing important details and clarifications. The comparison between DrAC-TR and DrAC isn’t fair since DrAC-TR essentially gets to train on many more levels of a game and because it sees the exact same levels in different themes it has the opportunity to learn theme invariance (perhaps even without the need to explicitly regularize the policy and value function).*
> >
> > **RESPONSE:** DrAC-TR use observations with same layout but different themes as augmentation (applied on top of default augmentations used by DrAC). While DrAC-TR trains on exactly the same number of levels, it does get to see the exact levels in different themes (as a result of augmentation) and hence is able to learn the theme invariance. However, our intent isn't to propose DrAC-TR as an alternative method. Rather, it is to use this theme randomization scheme, that makes use of privileged access to the game engine, as a proxy of better data augmentation schemes and argue that better data augmentation schemes can help close the generalization gap caused by appearance changes (i.e. theme variation).
> >
> > *It would be interesting to compare with PPO-TR to see how it compares to DrAC.*
> >
> > **RESPONSE:** PPO-TR gives final mean normalized score (MNS) of $0.36\pm0.12$ which is identical to UCB-DrAC ($0.37\pm0.1$). This is because UCB-DrAC has access to augmentations (like crop, rotate) which are independent of theme but helps in learning the policy. In contrast, PPO-TR only uses theme randomization as their augmentation and hence suffer in learning of policies. However, DrAC-TR has access to both (i) theme randomization and (ii) theme independent augmentations (such as crop, rotate). While (i) tries to make DrAC-TR independent of themes, (ii) helps DrAC-TR in learning the policy.
> >
> > *You can also apply DrAC but with augmentations given by different themes of the game.*
> >
> > **RESPONSE:** We are assuming that the reviewer wants to use different themes of the game as additional augmentations. This is precisely DrAC-TR which uses different themes as augmentations on top of the data augmentation used by DrAC.
> >
> > *An interesting question is whether you need to explicitly regularize the policy and value when you have this type of data and perhaps how many themes do you need to capture this invariance (to any other theme)?*
> >
> > **RESPONSE:** The base algorithm in DrAC-TR is UCB-DrAC which explicitly regularizes both the policy and the value.
> >
> > We agree that it will be interesting to study how many different themes do we need to achieve theme invariance in learned policy and we hope to include this experiment in the final version of the paper. In the current version of paper, we randomize over the entire set of procgen themes (i.e. DrAC-TR) to achieve theme invariance.
> >
> > *In Figure 2, the aggregated results don't seem to differ very much. However, there are games where the differences are significant, and others where they are not. I don't think the text reflects this very well and some of the claims can be misleading.*
> >
> > **RESPONSE:** The final test mean normalized score (MNS) of UCB-DrAC, DrAC-FT and DrAC-TR are $0.37\pm0.1$, $0.59\pm0.1$, and $0.51\pm0.11$ respectively. Hence, DrAC-FT is 59.4% better than DrAC and DrAC-TR is 37.8% better than DrAC. While UCB-DrAC, DrAC-FT, and DrAC-TR have similar performances on a subset of games, their performances differ when averaged across all games.
> >
> > *Can you provide some intuition on why DrAC-FT would be better than DrAC? Are they being trained and tested on the exact same dataset?*
> >
> > **RESPONSE:** Both DrAC and DrAC-FT are being trained and tested on different sets of levels.
> >
> > DrAC-FT refers to a setting in which we train DrAC over procgen games with fixed themes. Hence, both the test levels and the train levels in DrAC-FT share the same theme and only differ in layouts.
> >
> > In contrast, DrAC is trained on default setting of procgen games where each level differ in both theme and layout.
> >
> > DrAC-FT is better than DrAC because it doesn't have to learn to be invariant to themes as it is being trained and test on same theme. If DrAC were to learn theme invariant policies on default settings of procgen (i.e. different theme and different layout), it's final performance would be similar to that of DrAC-FT (which is theme invariant by design). However, DrAC isn't able to achieve theme invariance and hence is worse than DrAC.

---

> > > ### Author Response · Authors · 2021-11-23
> > > **Author's reply (3/3)**
> > >
> > > *While your experiments seem to suggest that the methods still capture level specific features, while having good generalization, and thus those features are needed to learn generalizable policies, I am not sure the claim is correct. There are level specific features which you need in order to take reasonable actions and others which you do not (so you want to be invariant to these). Perhaps a better test is whether the theme can be predicted (instead of the level), which is typically not useful for solving the task. In contrast, the layout typically is needed to both solve the task and predict the level of the game.*
> > >
> > > **RESPONSE:** We agree with the reviewer that there are certain level specific features (such as layout) which are needed to solve the task while other level specific features (such as theme) can be ignored. In fact, in section 4 (first paragraph), we also argue that some level specific features such as layout would be needed by policy to perform the right action and hence discouraging the policy from learning level specific features (as suggested by (Raileanu et al., 2021)) might not be the right thing to do. Finally, we show that policies that generalize still contain level specific information as they need to capture task relevant level specific features such as layout.
> > >
> > > We do agree that policies, that generalize well, will be less successful in predicting theme information as they will avoid capturing task irrelevant level specific features such as theme. However, this experiment is hard to conduct in procgen as each level has a unique theme.
> > >
> > > *It also depends how different the test set is from the train set. If you collected this using a random policy you might end up with essentially identical observations in the train and test set and thus it will be very easy to predict the level (due to memorization). I suggest collected the dataset using a pretrained policy or checking that the test set is completely different from the train set.*
> > >
> > > **RESPONSE:** We also did the experiment with expert trajectories (generated by a trained PPO policy on levels 1-200) and we got similar results. Hence, the results are not sensitive to the specific method for generating trajectories.
> > >
> > > **References:**
> > >
> > > [1] Rachit Dubey, Pulkit Agrawal, Deepak Pathak, Thomas L. Griffiths, and Alexei A. Efros. "Investigating Human Priors for Playing Video Games." ICML (2018).
> > >
> > > [2] Roberta Raileanu and Rob Fergus. "Decoupling Value and Policy for Generalization in Reinforcement Learning." ICML (2021).

---

> > > > ### Comment · Reviewer_185P · 2021-11-25
> > > > **Thank you for the response**
> > > >
> > > > I thank their authors for their comprehensive response to my comments. While some issues have been clarified, I still think some of the claims are not well-supported by the empirical evidence, some of the experiments are not very well designed to answer the posed questions, and overall it's unclear how general some of the conclusions are.
> > > >
> > > > In particular, I still don't think we can confidently draw a conclusion regarding "theme-invariance" using the experiment in the paper and I was disappointed that the authors didn't attempt to include the experiment I proposed to answer this question. I don't understand the authors point that "However, given procgen is deterministic, training (and testing) on a single layout would lead to policy memorizing an optimal sequence of the actions (using some spurious features) rather than learning theme invariant features." While the statement is correct, that is exactly what you want to probe -- whether the policy has learned theme-invariant features or simply memorized a sequence of actions that works well on the training environment but changes when the theme changes due to spurious correlations (the agent's policy is paying attention to the theme rather than the layout, and since that is different, the policy will likely change to being sub-optimal or even fail catastrophically if it's not robust).  Testing the policy on the environments with the same layout as the ones from training but different themes, would tell you to what degree the learned policy is theme-invariant.
> > > >
> > > > Overall, I think this paper is the beginning of some interesting investigations. However, I am not convinced the experiments described in this paper support the claims made by the authors. In addition, I think it would be better to focus on a one or two questions, clearly state them, and try to answer them as clearly as possible and provide as much insight and depth as possible. In its current form, it is not clear what questions the authors are trying to answer and what the conclusions are.
> > > >
> > > > Given all the above, I do not think the paper is ready for publication. However, I do think the subject of the paper is important and would be of interest for the community. I encourage the authors to take the feedback into account in order to improve the paper, and resubmit at a future conference.
> > > >
> > > > I also agree with the points made by other reviewers related to the fact that 1) the paper could benefit from analyzing a wider range of methods and providing more in-depth discussions, and 2) many of the observations made in this paper are not new, so I suggest the authors focus on questions the community doesn't yet have answers for.
> > > >
> > > > Finally, as mentioned in my original review, I think the types of questions the authors are asking are important, but the experimental setup and writing needs more work to warrant acceptance at ICLR. I suggest 1) better refining the questions they want to answer, 2) focus on a small number of questions (ideally one or two) and go into more depth to answer them (evaluating multiple methods, on multiple settings, using multiple experiments to probe various aspects etc.), 3) carefully design the experiments to more directly answer the questions you pose, 4) discuss the implications of the results in a more clear manner and ensure the conclusions are supported by the experiments, 5) openly discuss the limitations of the study and when the conclusions may not hold or the experiments are inconclusive. As mentioned before, Procgen contains many different games with very different settings (fully and partially observed, sparse and dense reward etc.). I expect some of these conclusions may differ from one setting to another, so disentangling these would lead to a much more valuable contribution.

---

### Official Review · Reviewer_sNjW · 2021-11-03

**Correctness:** 3
**Technical Novelty And Significance:** 2
**Empirical Novelty And Significance:** 3
**Recommendation:** 3
**Confidence:** 5

**Main Review:**

This paper proposes an empirical study of different mechanisms often set in place to prevent overfitting in Deep RL.
Although the empirical findings are interesting and well-conducted, they don't fully cover the span of regularization methods. More importantly, the authors point out the challenges and weaknesses but do not address them, limiting the impact of the contribution.

## Writing and clarity

This article appears to have been written in a hurry, without any proof-reading or work on the structure.
There are lots of typos, many articles or words missing, acronyms are undefined, one bibliographical reference is replaced with a question mark, the ordering of the bibliography is shady, etc.
Seeing the related work section after the conclusion is an unorthodox practice at best, and looks more as if it was written and just dropped there. Additionally, it is mostly redundant with the introduction section and most citations already appear in the paper.
Many ideas are repeated several times (e.g. the last paragraph of the introduction just repeats the previous ones, Section 8 is a repetition of the main take-away messages from Sections 3 to 7).
Despite these aspects and the fact that it lacks content (see next comments), the paper is rather clear, well-organized and pleasant to read.

## Lack of perspective on regularization for RL

The idea of preventing overfitting and fostering generalization is often tackled as regularization. As the authors point out, many recent work have concentrated on this aspect, including the three papers by Raileanu et al, which serve as a basis for this work. iDAAC being somewhat state-of-the-art, it is justified, but other approaches such as IBAC-SNI, random convolutions, or the "historical" RAD algorithm would be legitimate candidates to verify the empirical study conducted here. Additionally, as noted in (Song et al, 2019) regularization can be separated as explicit regularization (explicit penalization of the empirical risk) or implicit regularization on the architecture and optimizer (dropout, batchnorm, momentum, early stopping) or on the data (data augmentation). This categorization applies both to supervised and to reinforcement learning. The only place where regularization is covered as a whole is the very last paragraph of the paper. And a natural question is then to assess whether the techniques evaluated in the paper actually cover a significant span of the possible regularization techniques overall.

## Insufficient discussion

Most take-away messages take the form of a challenge and some perspectives for future work. Besides the raw results, the reader does not get much insight from the reading.

In Section 4, I think the discussion is very incomplete. It seems to me that it is necessary for the policy to rely on level-specific features and that, ideally, an algorithm that generalizes well should disentangle (I didn't see any reference to the topic of disentanglement in the paper) the theme information from the layout information and only base the policy on the latter. Therefore, I think the problem should not be to evaluate whether "policies have level-specific features" but rather to assess whether these features are theme-independent. the discussion on this question is too shallow (almost inexistent).
More anecdotically, the claim that "policies have level-specific features regardless of whether it generalizes or not" is not backed up by Table 1 since the policies that have been used in this table "have good generalization capabilities".

In Section 5, the discussion on auxiliary tasks deserves a lot more details. The presented results display a lot of variability and the actual benefit of the proposed auxiliary task is very dependent on the game tackled. This section would benefit a lot from discussing the reasons of this variability. For example, auxiliary tasks could be expected to bring in game-dependent benefits since it depends on whether the auxiliary tasks features help disambiguate between equivalent sets of features for the policy. Unfortunately, the discussion stops after the plain presentation of results.
Also, data augmentation was mixed up with domain randomization. I assume it is a moment of inattention coupled with a lack of proof-reading.

Section 6 (unfortunately) is nothing new. Hyperparameter sensitivity is (unfortunately) a recurrent, known problem in RL (even without the problem of generalization). And (unfortunately), no new insights are provided in this section.

Figure 6 (Section 7) deserves a much longer discussion. Why does training on more levels hinder the overall performance? Is the training done sequentially or in parallel (in the latter case, is it only a matter of number of samples to process before moving on to a next step?)? Even the raw results are not sufficient here to inform the reader on the mechanisms at hand.

**Summary Of The Paper:**

This paper proposes an empirical study of different mechanisms often set in place to prevent overfitting in Deep RL. It draws conclusions at what the challenges are and suggests future work directions.

**Summary Of The Review:**

This paper proposes an empirical study of different mechanisms often set in place to prevent overfitting in Deep RL.
Although the empirical findings are interesting and well-conducted, they don't fully cover the span of regularization methods. More importantly, the authors point out the challenges and weaknesses but the discussion is very shallow and they do not address them, limiting the impact of the contribution.
The paper itself needs an important effort of editing (it is still a very preliminary version, assembled in a rush), and a significant contribution to improve the generalization capability of RL algorithms, to have a meaningful impact for the community.

---

> ### Author Response · Authors · 2021-11-23
> **Author's reply (1/2)**
>
> We would like to thank the reviewer for their constructive feedback. We address specific points below:
>
> *The idea of preventing overfitting and fostering generalization is often tackled as regularization. As the authors point out, many recent work have concentrated on this aspect, including the three papers by Raileanu et al, which serve as a basis for this work. iDAAC being somewhat state-of-the-art, it is justified, but other approaches such as IBAC-SNI, random convolutions, or the "historical" RAD algorithm would be legitimate candidates to verify the empirical study conducted here. Additionally, as noted in (Song et al, 2019) regularization can be separated as explicit regularization (explicit penalization of the empirical risk) or implicit regularization on the architecture and optimizer (dropout, batchnorm, momentum, early stopping) or on the data (data augmentation). This categorization applies both to supervised and to reinforcement learning. The only place where regularization is covered as a whole is the very last paragraph of the paper. And a natural question is then to assess whether the techniques evaluated in the paper actually cover a significant span of the possible regularization techniques overall.*
>
> **RESPONSE:** We believe that our analysis covers a substantial span of regularization techniques that have been attempted in the Deep RL literature. We agree with the reviewer that we have not compared against IBAC-SNI, RAD or implicit regularization methods. The reason is that these methods are worse than UCB-DrAC (which we study in Section 3) as detailed below:
>
> - Random Convolution is one of many data augmentation techniques used by UCB-DrAC.
> - Similarly, RAD is another data-augmentation algorithm that is very similar to DrQ (on which UCB-DrAC is built). While DrQ takes an average over multiple data augmentations for calculating target value, RAD only uses a single data augmentation to calculate target value.
>
> We didn't analyze IBAC-SNI as it only outperforms PPO in Coinrun. As shown by Raileanu et al. (2021), when tested across procgen games, the median performance is worse than PPO.
>
> Dropout is a specific form of noise injection which is included in IBAC-SNI. Early stopping won't help with generalization in procgen games given we plot test performance (along with training performance) every few training epochs and we don't see any dip in test performance in the learning curves.
>
> We didn't analyze batchnorm as it offers minimal gains in test performance of procgen games, as shown by Wang et al., 2020.
>
> *In Section 4, I think the discussion is very incomplete. It seems to me that it is necessary for the policy to rely on level-specific features and that, ideally, an algorithm that generalizes well should disentangle (I didn't see any reference to the topic of disentanglement in the paper) the theme information from the layout information and only base the policy on the latter. Therefore, I think the problem should not be to evaluate whether "policies have level-specific features" but rather to assess whether these features are theme-independent. the discussion on this question is too shallow (almost inexistent).*
>
> **RESPONSE:** We agree with the reviewer's assessment that there are certain level-specific features (such as layout) that are needed to solve the task, while other level-specific features (such as theme) can be ignored. In fact, in section 4 (first paragraph), we argue that some level-specific features such as layout would be needed by policy to perform the right action. However, recent works (Raileanu et al., 2021) have argued to discourage the policy from learning level-specific features. We wanted to argue that this insight is incomplete and the policies, that generalize well, still need to learn task-relevant level-specific features such as layout.
>
> *More anecdotically, the claim that "policies have level-specific features regardless of whether it generalizes or not" is not backed up by Table 1 since the policies that have been used in this table "have good generalization capabilities".*
>
> **RESPONSE:** There might be a misunderstanding. The statement in the paper is:
>
> "To answer this, we first take the policies trained by PPO on 100k levels, by DAAC on 200 levels and by iDAAC on 200 levels on different games of procgen. All these policies have good generalization capabilities."
>
> We should point out that the only policy that does not have good generalization performance in Table 1 is PPO (200): PPO trained on 200 levels.  Please let us know if we have misunderstood you or something else is unclear.

---

> > ### Author Response · Authors · 2021-11-23
> > **Author's reply (2/2)**
> >
> > *In Section 5, the discussion on auxiliary tasks deserves a lot more details. The presented results display a lot of variability and the actual benefit of the proposed auxiliary task is very dependent on the game tackled. This section would benefit a lot from discussing the reasons of this variability. For example, auxiliary tasks could be expected to bring in game-dependent benefits since it depends on whether the auxiliary tasks features help disambiguate between equivalent sets of features for the policy. Unfortunately, the discussion stops after the plain presentation of results.*
> >
> > **RESPONSE:** We would have loved to have provided such an analysis. However, despite our best attempt we were unable to distill how auxiliary losses help in terms of simple explanations. Figure 7 (appendix) groups procgen games by methods which perform the best on those games. However, because we didn't find consistent patterns, we did not include them in the main text.
> >
> > The main reason for including this section is to point out that simple auxiliary losses outperform data-augmentation methods used in practice today.
> >
> > *Also, data augmentation was mixed up with domain randomization. I assume it is a moment of inattention coupled with a lack of proof-reading.*
> >
> > **RESPONSE:** Domain randomization (theme randomization) can also be viewed as a form of data augmentation. We intended to use the theme randomization scheme (DrAC-TR), that makes use of privileged access to the game engine, as a proxy for better data augmentation schemes to investigate if better data augmentation schemes can help close the generalization gap.
> >
> > *Section 6 (unfortunately) is nothing new. Hyperparameter sensitivity is (unfortunately) a recurrent, known problem in RL (even without the problem of generalization). And (unfortunately), no new insights are provided in this section.*
> >
> > **RESPONSE:** While it is expected that using game-specific hyper-parameters leads to improved performance, it does raise the question if the improvement in performance of new methods comes from careful hyper-parameter tuning or from the method itself. Moreover, the baseline PPO uses same hyperparameters across all games.
> >
> > *Figure 6 (Section 7) deserves a much longer discussion. Why does training on more levels hinder the overall performance?*
> >
> > **RESPONSE:** We don't claim that training on more levels hinders overall performance. Rather, the strategy of training a policy on training levels (1-200) and then fine-tuning only the policy head-on new levels leads to reduced performance on new levels as the number of new levels increase. This is due to the limited capacity of shallow (2 layer) policy head which is able to adapt the frozen visual features (learned on levels 1-200) to 200 and 1k new test levels but not to 10k and 100k new test levels. We have updated the draft to clarify this point in Section 7.
> >
> > *Is the training done sequentially or in parallel (in the latter case, is it only a matter of number of samples to process before moving on to a next step?)? Even the raw results are not sufficient here to inform the reader on the mechanisms at hand.*
> >
> > **RESPONSE:** We first completely train a policy on training levels 1-200 and then fine-tune the policy head (or the full policy) on new levels (where a number of new levels can be 200, 1k,  10k, or 100k). We will be happy to clarify more as we weren't quite sure what the reviewer mean by training being done sequentially or in parallel.
> >
> > *This article appears to have been written in a hurry, without any proof-reading or work on the structure. There are lots of typos, many articles or words missing, acronyms are undefined, one bibliographical reference is replaced with a question mark, the ordering of the bibliography is shady, etc. Seeing the related work section after the conclusion is an unorthodox practice at best, and looks more as if it was written and just dropped there. Additionally, it is mostly redundant with the introduction section and most citations already appear in the paper. Many ideas are repeated several times (e.g. the last paragraph of the introduction just repeats the previous ones, Section 8 is a repetition of the main take-away messages from Sections 3 to 7). Despite these aspects and the fact that it lacks content (see next comments), the paper is rather clear, well-organized and pleasant to read.*
> >
> > **RESPONSE:** We have tried to fix the typos in the paper. The related work section appears after the introduction.
> >
> > **REFERENCES**
> >
> > [1] Roberta Raileanu and Rob Fergus. "Decoupling Value and Policy for Generalization in Reinforcement Learning." ICML (2021).
> >
> > [2] Kaixin Wang, Bingyi Kang, Jie Shao, Jiashi Feng. "Improving Generalization in Reinforcement Learning with Mixture Regularization." NeurIPS (2020)

---

> > > ### Comment · Reviewer_sNjW · 2021-11-28
> > > **Feedback**
> > >
> > > I thank the authors for their responses on my comments and I apologize for answering a bit late.
> > > I still think the paper deals with a very important issue, but the discussion remains too shallow and the contribution is not strong enough for publication.
> > > I'm keeping my score.

---

### Decision · Program_Chairs · 2022-01-20

**Decision:**

Reject

**Comment:**

This paper presents an empirical study of generalization in visual reinforcement learning. This study is carried out in the domain of video games and it addresses the benefits of techniques such as regularization, augmentation and training with auxiliary tasks. The reviewers for this submission were positive about the goal and setups in this paper. They agreed that understanding why present day methods that attempt to improve generalization continue to fall short, is an important problem. However, most reviewers were underwhelmed by the findings presented in the submission. As examples: Reviewer 185P mentions that "the paper does not seem to provide a clear and definite answer to the question" and " I am not convinced the experiments described in this paper support the claims made by the authors." and Reviewer SFef mentions that "Most of the conclusions are already known". Some reviewers also found a lack of clarity and several typos in the initial submission. The authors have provided detailed responses to the reviewers. In particular they have fixed most writing issues. They also detailed why certain algorithms and techniques were benchmarked in this submission and others were left out. I think this is reasonable. One cannot expect a paper to benchmark every algorithm out there, and choosing promising and representative ones is sufficient. My takeaway from detailed discussions about this paper are that: The paper is much improved from a writing point of view and the rebuttal addresses some concerns well. However, I do agree with the reviewers that the findings presented in the paper are for the most part expected. This reduces the value of the paper to readers. When this is the case, it may be beneficial to dig deeper into these findings and present a narrow but deep analysis. Please see Reviewer 185P's suggestions in this regard. Given the above, I am recommending rejection for this conference, but I encourage the authors to take into the reviewers suggestions and resubmit.